# GABARAP proteins regulate the packaging of HIV-1 genomic RNA into virions

Marjory Palaric [ID][1], Margaux Versapuech[1], Delphine Judith[1], Corentin Aubé[1], Marjorie Leduc [ID][2], Jacques Dutrieux [ID][1], Emilie-Fleur Gautier[2], Jean-Christophe Paillart [ID][3], Sarah Gallois-Montbrun [ID][1] & Clarisse Berlioz-Torrent [ID][1✉]

## Abstract

In addition to their role in canonical autophagy, autophagy proteins (ATG) contribute to various cellular processes, including phagocytosis, membrane remodeling, and vesicle secretion. Several viruses also exploit components of the autophagy pathway for their own replication. Here, we explore the role of ATG proteins in HIV-1 assembly. Postulating that host proteins crucial for virion assembly are present at the assembly site and can be incorporated within virions, we analyze the proteome of HIV-1 preparations using mass spectrometry. We identify an enrichment of macroautophagy-related terms, notably 3 of the 6 ATG8 (LC3/GABARAP) proteins. Functional studies reveal that GABARAP proteins are critical for the production of infectious virions. Knockout of GABARAP proteins reduces the packaging of viral genomic RNA (gRNA) into particles, impairing virion infectivity. GABARAPL1 associates with gRNA and interacts with Gag in an RNA-dependent manner. Additionally, GABARAP knockout increases cellular Gag:gRNA complexes and decreases gRNA association with membranes, suggesting that GABARAP proteins regulate gRNA fate during HIV-1 assembly by facilitating its packaging. This study uncovers a novel role for GABARAP proteins in HIV-1 genome packaging.

**Keywords** HIV-1; RNA; GABARAP; Autophagy Protein; Genome Packaging
**Subject Categories** Autophagy & Cell Death; Membranes & Trafficking; Microbiology, Virology & Host Pathogen Interaction

## Introduction

The ATG (AuTophaGy-related genes) proteins is a group of proteins that control the autophagy process, an evolutionarily conserved degradative pathway essential for maintaining cellular homeostasis, responding to pathological processes and fighting infections (Klionsky et al, 2021; Mizushima, 2018). Among the ATG proteins, ATG8 proteins are ubiquitin-like proteins essential for autophagy, involved in autophagosome membrane expansion, cargo recruitment, and autophagosome–lysosome fusion (Johansen and Lamark, 2020). In humans, this family includes six members, LC3A, LC3B, LC3C, GABARAP, GABARAPL1, and GABARAPL2, divided in mammals in two subfamilies, LC3 (microtubule-associated proteins 1A/1B light chain 3) and GABARAP (γ-aminobutyric acid receptor-associated protein). Structurally, all ATG8s share a ubiquitin-like core and N-terminal α-helices enabling membrane tethering and protein interactions. LC3 proteins mainly mediate cargo recognition and transport, while GABARAPs are crucial for autophagosome maturation and fusion with lysosomes. Beyond autophagy, GABARAPs regulate membrane trafficking and neuronal development (Wang and Olsen, 2000; Mansuy et al, 2004). During autophagy, ATG8s undergo covalent conjugation to phospholipids (lipidation), classically phosphatidylethanolamine (PE), within the membrane of the forming autophagosome. Lipidated forms of ATG8 (denoted ATG8-II) are anchored to the phagophore, interact with cargoes to be degraded or with autophagy receptors during "selective autophagy" (Lystad and Simonsen, 2019) and recruit them to the inner membrane of the phagophore (Johansen and Lamark, 2020).

Over the last decade, growing evidence has demonstrated that ATG proteins also play roles in various non-autophagy-related cellular functions linked to membrane biology, such as endocytosis, phagocytosis, secretion and exocytosis (Galluzzi and Green, 2019). Additionally, they are involved in membrane-independent functions, including immune and inflammatory responses and the maintenance of genomic stability. For instance, certain ATG proteins regulate a subset of non-canonical autophagy pathways known as CASM (conjugation of ATG8 to single membranes), which promote lysosomal degradation or recycling by targeting ATG8 proteins to preexisting single-membrane structures (Durgan and Florey, 2022; Figueras-Novoa et al, 2024). Recent advances also demonstrate that components of the autophagy machinery specify the loading and secretion of RNA-binding proteins and RNA into extracellular vesicles (Leidal et al, 2020). The non-autophagic functions of the autophagy machinery are also hijacked by viruses for their own benefit (Münz, 2021). The subversion of components of the autophagy machinery is a common mechanism shared by multiple RNA viruses, including viruses from the *Flaviviridae* family, Coronaviruses and Influenza A virus (Jassey and Jackson, 2024). Several studies notably highlight that enveloped viruses have

[1]Université Paris Cité, CNRS, Inserm, Institut Cochin, F-75014 Paris, France. [2]Proteom'IC facility, Université Paris Cité, CNRS, Inserm, Institut Cochin, F-75014 Paris, France. [3]Université de Strasbourg, CNRS, Architecture et Réactivité de l'ARN, UPR 9002, Strasbourg, France. ✉E-mail: clarisse.berlioz@inserm.fr

evolved mechanisms to mobilize autophagy resources, ATG proteins, and autophagic vesicles from host cells for viral replication, viral membrane production and viral release (Beale et al, 2014; Abernathy et al, 2019; Reggiori et al, 2010). Some RNA viruses, such as Poliovirus and Coxsackievirus B, hijack secretory autophagy to facilitate their release without lysis of the host cell. Virions are recruited to autophagosome-like structures for release into the extracellular environment in LC3-decorated microvesicles (Dahmane et al, 2022; Jassey and Jackson, 2024). These double membrane vesicles are important for the replication and assembly of these naked viruses (Dahmane et al, 2022; Jackson et al, 2005; Richards and Jackson, 2012), but also for certain enveloped viruses such as the *Flaviviridae* (Panyasrivanit et al, 2009). LC3/GABARAP proteins are also involved in the budding of Influenza A virus (IAV) (Beale et al, 2014) and in the assembly and release of certain DNA viruses, such as Herpesviruses (Epstein-Barr virus (EBV), Varicella zoster virus (VZV) and human cytomegalovirus (HCMV)) where LC3B-II and GABARAPL1-II are found associated with viral particles and whose presence is implicated in viral envelope acquisition (Buckingham et al, 2016; Nowag et al, 2014; Taisne et al, 2019). In the case of HIV-1 infection, few studies have investigated the involvement of ATG proteins in the late stages of the replication cycle. Our laboratory has shown that HIV-1 Vpu protein antagonizes BST2 antiviral functions *via* multiple mechanisms, including the subversion of an LC3C-associated pathway, a key cell-intrinsic antimicrobial mechanism (Madjo et al, 2016; Judith et al, 2023). Likewise, the HIV-1 Vif protein has been shown to bind to LC3B and decrease autophagy, independently of its restrictive function on APOBEC3G (Borel et al, 2015). ATG9 has also been reported to contribute to HIV-1 infectivity (Mailler et al, 2019). However, no further studies have explored the possible contribution of LC3/GABARAP proteins in HIV-1 assembly.

HIV-1 particle assembly begins in the cytosol and ends at the plasma membrane following the initiation of Pr55Gag polyprotein (Gag) multimerization and its interaction with viral genomic RNA (gRNA) (Bernacchi, 2022; Freed, 2015; Sumner and Ono, 2024). This assembly leads to the packaging of a dimeric viral gRNA into nascent virions (Rein, 2019; Hanson et al, 2022). This step requires the integrity of the matrix (MA), capsid (CA) and nucleocapsid (NC) domains of Gag (Kutluay et al, 2014; Lei et al, 2023; Lerner et al, 2022; Thornhill et al, 2019), and two regions present in the 5' untranslated region (5' UTR) of HIV-1 gRNA, (i) the SL1-SL3 region that interacts with Gag and contains the gRNA dimerization initiation site (DIS/SL1) and the main ψ packaging sequence (SL3) (Abd El-Wahab et al, 2014; Comas-Garcia et al, 2018; Dilley et al, 2017; Rein, 2019; Skripkin et al, 1994) and (ii) the polyadenylation signal (PAS: $_{73}$AAUAAA$_{78}$) present in the 5' end of the viral genome (Didierlaurent et al, 2011; Nikolaitchik et al, 2021; Smyth et al, 2018; Yasin et al, 2024). HIV-1 gRNA results from the RNA polymerase II-dependent transcription of the integrated HIV-1 provirus and the export of a full-length unspliced HIV-1 RNA. Besides its function as a viral genome, the unspliced HIV-1 RNA is translated to generate the Gag and Gag-Pol precursors, orchestrating the viral assembly. It also undergoes complex alternative splicing events to generate partially and completely spliced HIV-1 RNA species, leading to the translation of Env glycoproteins, and the accessory and regulatory proteins (Nguyen Quang et al, 2020). Interestingly, recent advances in the field showed that differential transcriptional start site (TSS) usage modulates the number of

guanosines (Gs) at the 5'-end of the RNA genome leading to two major HIV-1 RNA structures either competent for gRNA dimerization and packaging (1 G gRNA) or translation (2Gs but mainly 3Gs mRNA) (Brown et al, 2020; Ding et al, 2021; Duchon and Hu, 2024; Kharytonchyk et al, 2016; Nikolaitchik et al, 2023; Yasin et al, 2024). Despite many studies aimed to gain a better understanding of HIV-1 gRNA packaging, the cellular players regulating this process remain poorly understood.

Considering the works describing the non-autophagic role of ATG proteins in the assembly, morphogenesis, and release of various viruses, we examined whether ATG proteins are involved in the assembly of infectious HIV-1 particles. We hypothesized that host proteins essential for viral assembly are located at the assembly site and can be incorporated into virions. Using an unbiased mass spectrometry approach, we identified the proteome of HIV-1 preparations. Gene ontology analysis showed an enrichment of terms related to translation, protein catabolism, intracellular trafficking, cytoskeleton organization and regulation of macro-autophagy. We found that three of six ATG8 family members were significantly enriched in the viral preparation and characterized their incorporation into virions. Using CRISPR-Cas9, we revealed that GABARAP subfamily members are necessary for producing infectious virions. Knockout of GABARAP proteins caused an increased release of viral particles exhibiting defective gRNA packaging, along with an accumulation of viral gRNA in the cytoplasm, which remained mainly spatially separated from the plasma membrane. Our findings revealed the critical role of GABARAP proteins in gRNA packaging into virions.

## Results

### Proteomic analysis of HIV-1 preparations

Given recent works showing that various viruses are capable of mobilizing some components of the autophagy machinery for their morphogenesis and release, we investigated whether ATG proteins participate in the formation of infectious HIV-1 particles. We postulated that host proteins crucial for viral particle morphogenesis are present at the assembly site and are susceptible to be incorporated within virions, and we thus established a proteomic analysis of virion-containing supernatants of HIV-1-producing cells. To this end, HeLa cells were transfected with HIV-1 NL4-3 provirus and, 48 h after transfection, culture supernatants of HIV-1 producer cells were collected, filtered and ultracentrifuged on a 20% sucrose cushion to pellet virions released in the supernatants (Fig. 1A). As a control, supernatant from non-transfected HeLa cells was collected and prepared as the supernatant of HIV-1 producer cells (mock preparation). The same volume of filtered and ultracentrifuged supernatants from mock- and HIV-1-producing cells were then subjected to LC-MS analysis using a TimsTOFPro in data-independent acquisition (DIA) mode and a label-free quantification method, a powerful and highly sensitive proteomics method that allows the quantification of a large number of peptides and proteins. The acquired data were analyzed using DIA-NN software. Interrogation of the HIV-1 database that includes all viral proteins showed that Gag, Pol and Env proteins are highly enriched in viral preparation (Fig. 1B; Dataset EV1). Small accessory Nef, Vpr, and Vif proteins were also found enriched, as previously

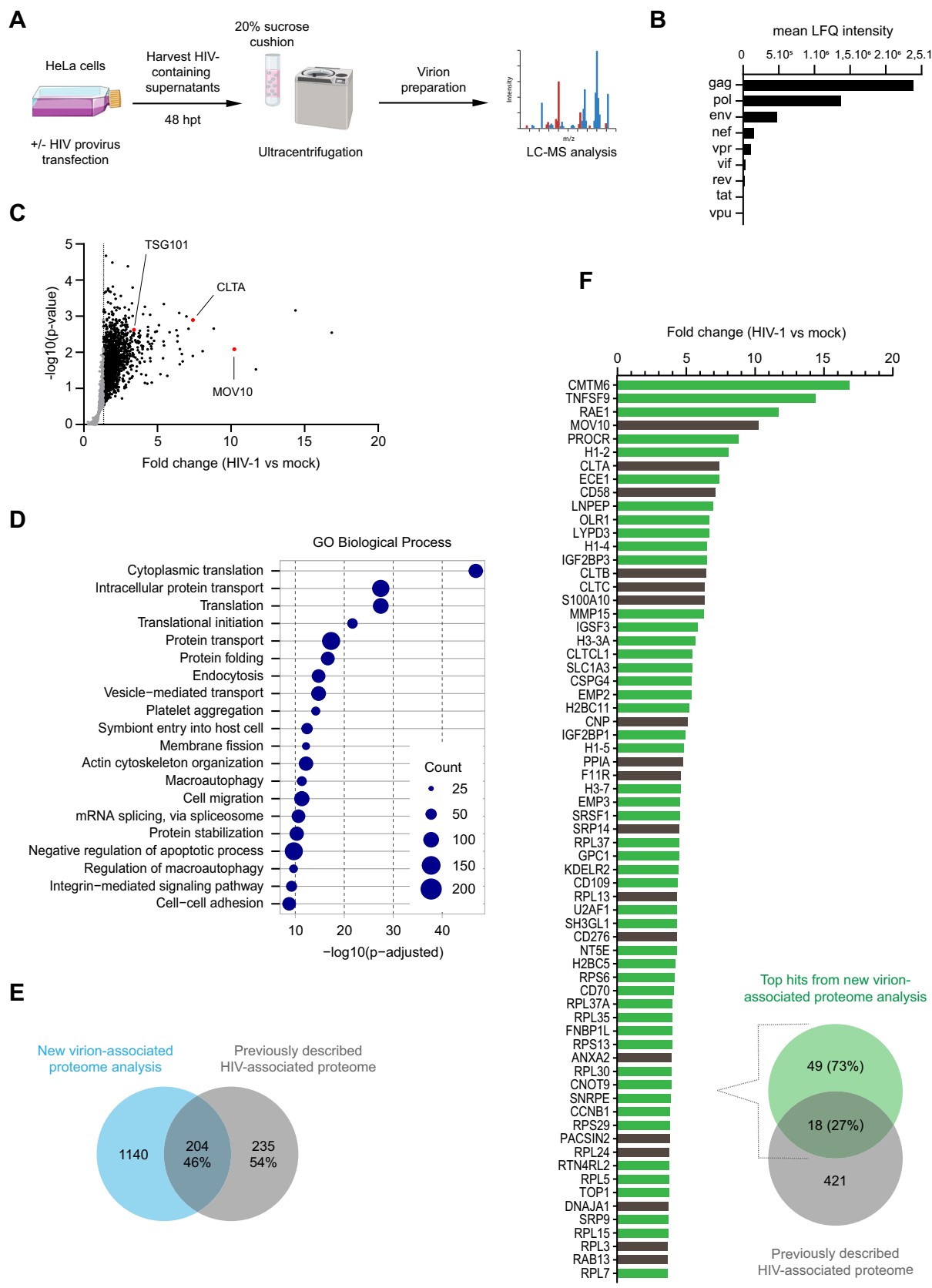

◄  **Figure 1.  Proteomic analysis of HIV-1 preparations.**

(A) Outline of HIV-1 virion preparation for LC-MS analysis: HeLa cells were transfected with WT NL4-3 HIV-1 proviral DNA or non-transfected as a control for 48 h. Supernatants were collected, filtered and ultracentrifuged on a 20% sucrose cushion. Virions-enriched preparations were subsequently lysed and analyzed by LC-MS. (B) LC-MS intensities (LFQ) of viral proteins identified by LC-MS in viral preparations from NL4-3 HIV-1 transfected HeLa cells as described in (A) ($n = 4$ biological replicates). (C). Cellular proteins identified by LC-MS in four viral preparations from NL4-3 HIV-1 transfected cells are shown in a scatter plot of fold change (HIV-1 *vs* mock) and $-\log 10$($p$ value). The proteins present in four viral preparations, and in 0, 1 or 2 mock preparations are not shown here because they are not statistically evaluable. Black and red dots indicate proteins that are enriched (FC ≥1.3) in viral preparation of NL4-3 HIV-1 transfected cells as compared to non-transfected cells. Red dots indicate some proteins that have been previously described in the HIV-associated proteome. (D) GO analysis (biological process) of cellular proteins enriched in four viral preparations from NL4-3 HIV-1 transfected cells. Enrichment of 2919 HIV-1 virion-associated proteins plotted as a function of $-\log 10$($p$ adjusted). The diameter of each blue circle correlates with the number of proteins (counts) associated with each term. (E) Diagram highlighting the number of cellular proteins significatively enriched (FC ≥1.3; $p < 0.05$) in NL4-3 HIV-1 virions purified as described in (A) ($n = 4$ biological replicates) (in blue), or in previously described HIV-associated proteome (in gray). (F) Top hits (5%) of cellular proteins significatively enriched (FC ≥1.3; $p < 0.05$) in the viral preparation from NL4-3 HIV-1 transfected cells analyzed by LC-MS (in green) as compared to previously described HIV-associated proteome (in gray). The histogram represents the fold change of the top hits of enriched cellular proteins identified by LC-MS (in green), including the ones in common with the previously described HIV-associated proteome (in gray) ($n = 4$ biological replicates). Data information: Statistical analysis was performed using a one-tailed paired Student $t$-test in (C) and Fisher's exact test with Benjamini–Hochberg correction in (D); $n = 4$ biological replicates. Source data are available online for this figure.

described (Bukovsky et al, 1997; Cohen et al, 1990; Liu et al, 1995). Importantly, and as expected, Vpu, Tat, and Rev proteins were present in a very limited quantity (Fig. 1B).

To identify host factors associated with virions, LC-MS data were then used to query a human protein database. In total, 2919 cellular factors were identified in four biological replicates in HIV-1 preparations (Dataset EV1), in which 1838 proteins were enriched with a fold change (FC) ≥1.3 (Fig. 1C; black and red dots) and 46 proteins were exclusively detected in HIV-1 preparations, compared to mock preparations (Fig. EV1A). Gene Ontology analysis of the 2919 proteins present in our four viral preparations revealed an enrichment for terms linked to translation, protein folding and catabolism, intracellular trafficking, cytoskeleton organization and regulation of macroautophagy (Fig. 1D).

We then focused on a set of proteins significantly enriched (FC ≥1.3 and $p < 0.05$) across all four viral preparations, comprising 1344 human proteins (Dataset EV1). Comparing this set of proteins with those previously identified as part of the HIV-1 virions associated proteome, collected from several previous studies (Fig. 1E) (Brégnard et al, 2013; Chertova et al, 2006; Gale et al, 2019; Ott et al, 2000; Santos et al, 2012; Saphire et al, 2006), we confirmed 204 (46%) of the 439 previously reported proteins. Moreover, among the 67 top hits (top 5%) of our 1344 significative enriched proteins in HIV-1 supernatants (Dataset EV1), 18 proteins were previously found enriched in HIV-1 virions (27%), such as the RNA-binding protein MOV10 and the clathrin proteins CLTA, CLTB, CLTC, and CD58 (Fig. 1F).

In addition, among the top 5% of enriched proteins, our study uncovered 49 novel proteins associated with HIV-1 virion preparations, enriching the database of preexisting HIV-1-associated proteome and highlighting the sensitivity and analysis depth of the DIA mode.

## LC3/GABARAP proteins are incorporated into HIV-1 particles

Among the enriched proteins present in Fig. 1D, we focused on the ones related to the autophagy process. They regroup 41 proteins (FC ≥1.3), which are present in all four HIV-1 preparations compared to preparations from mock cells. This list included proteins from the "macroautophagy" term of the GO analysis described above (Fig. 1D), and 12 additional proteins that are implicated in autophagy but were not associated with this term (Fig. 2A,B; Dataset EV1). Within this list of 41 enriched proteins

(FC ≥1.3), 18 are associated with the ESCRT pathway (in blue) and 23 with the autophagy machinery (purple and pink) (Fig. 2A,B; Dataset EV1). The ESCRT-I proteins TSG101, VPS28, and MVB12 A, the ESCRT-III proteins CHMP4A and CHMP4B, and the ESCRT-associated protein PDCD6IP/Alix were among the most enriched proteins (2 to 3.4-fold change). These proteins have previously been described as proviral factors for HIV-1 budding (Morita et al, 2011; Strack et al, 2003; von Schwedler et al, 2003; Carlton and Martin-Serrano, 2007), validating our experimental approach. Several autophagy proteins implicated in the initiation (ULK1, ATG101), the nucleation (ATG9A) and the elongation (ATG3, ATG7, and ATG16L1) steps of autophagy were also found to be enriched in the viral preparation. Interestingly, we also found four of the six proteins from the ATG8 family, MAP1LC3B (LC3B), GABARAPL1, GABARAP, and GABARAPL2. LC3B, GABARAPL1, and GABARAPL2 were found statistically enriched ($p < 0.05$) with a notable 2.3-, 2-, and 1.3-fold change, respectively (Fig. 2A,B; Dataset EV1). These fold change enrichments are similar to those found for the ESCRT-III CHMP4B protein and the ESCRT-associated protein PDCD6IP/Alix (2.2-fold and 2-fold change, respectively). We also noticed that four autophagy receptors (TAX1BP1, SQSTM1, NBR1, CALCOCO1), known to interact with ATG8 proteins, were also present and enriched in viral preparations. As described previously (Garcia-Moreno et al, 2023; Murigneux et al, 2024), we observed a modest correlation between the LC-MS signal intensity of cellular factors found in viral preparations and their abundance in HIV-1 provirus-transfected HeLa cells ($R = 0.34$; Fig. EV1B). This suggests that the presence of some proteins in viral supernatants may reflect their expression levels in producer cells, although this relationship appears limited. Notably, when considering the enrichment of these proteins in viral versus mock preparations, rather than their absolute LC-MS signal intensity, the correlation was essentially lost ($R = 0.016$; Fig. EV1C), indicating that the presence of proteins related to the autophagy pathway in viral preparations does not simply mirror their expression in producer cells. Instead, their selective enrichment (1.3≤ FC ≤3.41) may point to a specific association with the viral particles (Fig. EV1C).

Considering the abundance of ATG8 proteins in viral preparation, we next analyzed the association of these proteins with HIV-1 virions by western blotting. Culture supernatants of HIV-1 provirus-transfected HeLa cells (Fig. 2C) and of HIV-1-infected primary CD4 + T cells (Fig. 2D) were ultracentrifuged on a 20%

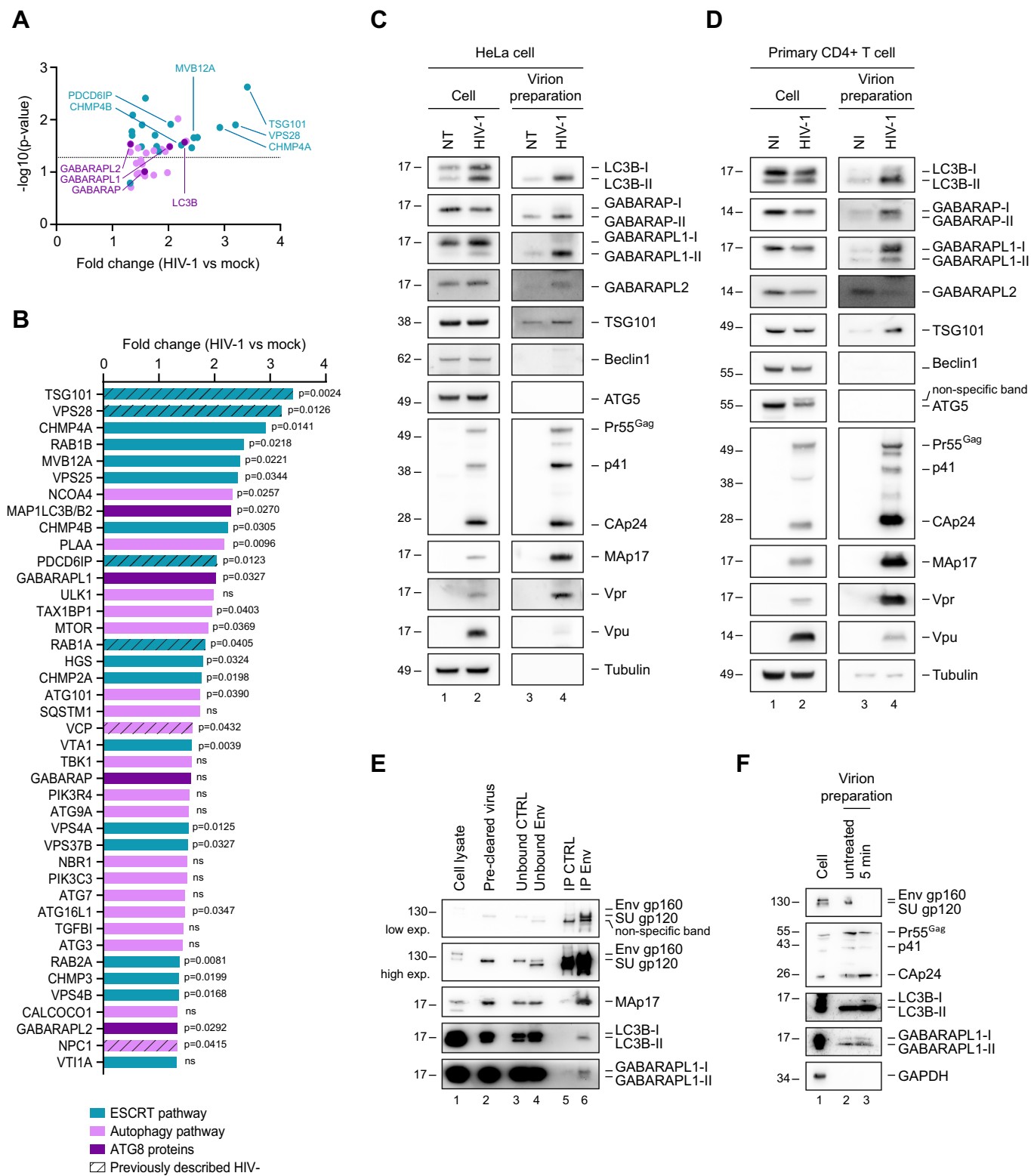

sucrose cushion and analyzed by western blot. Non-transfected (NT) and non-infected (NI) ultracentrifuged cell supernatants were used as controls. As expected, the HIV-1 structural proteins Pr55Gag (Gag) and its subproducts MACAp41, CAp24, and MAp17, that composed the architecture of the viral particle, were

detected exclusively in viral preparations and not in controls (Fig. 2C,D). We also observed a selective incorporation of the cellular ESCRT-I protein, TSG101. By contrast, the cellular ATG proteins, ATG5 and beclin1, and the HIV-1 Vpu protein were poorly detected in viral preparation (Fig. 2C,D), as in our LC-MS

**Figure 2. Association of ATG8 proteins with HIV-1 virions.**

(A) HeLa cells were transfected with WT NL4-3 HIV-1 proviral DNA or non-transfected as a control. Supernatants were collected 48 h later, filtered and ultracentrifuged on a 20% sucrose cushion. Virions-enriched preparations were subsequently lysed and analyzed by LC-MS. ATG proteins (41) identified by LC-MS analysis in preparation from mock and NL4-3 HIV-1 transfected HeLa cells, are shown in a scatter plot of fold change (HIV-1 vs mock) and −log10(p value) (the hatched line represents the p value = 0.05). Blue dots indicate proteins controlling the ESCRT pathway. Pink and purple dots indicate proteins controlling macroautophagy; ATG8 proteins are represented in purple. (B) Proteins described in (A) are shown in a histogram of fold change. The hatched proteins are the common ones with the previously described HIV-associated proteome. (C) HeLa cells were transfected or not (NT non-transfected) with WT NL4-3 HIV-1 proviral DNA. Supernatants were collected, filtered and ultracentrifuged on a 20% sucrose cushion 48 h later. Western blot analysis of LC3B, GABARAP, GABARAPL1, GABARAPL2, TSG101, Beclin1, ATG5, Gag and CAp24 products, MAp17, Vpr, Vpu and Tubulin, in producing cells and supernatants. (D) LT CD4+ primary cells were infected, or not (NI, non-infected), with VSVg pseudotyped NL4-3 HIV-1 for 48 h at a MOI of 0.5. Western blot analysis of LC3B, GABARAP, GABARAPL1, GABARAPL2, TSG101, Beclin1, ATG5, Gag, and CAp24 products, MAp17, Vpr, Vpu, and Tubulin, in producing cells and supernatants. (E) HeLa cells were infected with VSVg pseudotyped NL4-3 HIV-1 at an MOI of 1. Supernatants were collected 48 h later, filtered and ultracentrifuged. IP of HIV-1 viral particles using human monoclonal anti-SUgp120 HIV-1 or human IgG1k as control. Western blot analysis of HIV-1 SUgp120, HIV-1 MAp17, LC3B, and GABARAPL1. (F) HeLa cells were infected with VSVg pseudotyped NL4-3 HIV-1 at an MOI of 1. Cell lysates were recovered 48 h later, and supernatants were collected, filtered and ultracentrifuged on a 20% sucrose cushion. Control (untreated) and subtilisin-treated supernatants were lysed, and the following proteins were analyzed by western blot: HIV-1 SUgp120, HIV-1 Gag and CAp24 products, LC3B, GABARAPL1 and GAPDH. All western blots are representative of at least three biological replicates. In all Western blots, LC3B-I, GABARAP-I, and GABARAPL1-I correspond to the non-lipidated form of these proteins and LC3B-II, GABARAP-II, and GABARAPL1-II to the lipidated forms. Data information: Statistical analysis in (A, B) was performed using a one-tailed paired Student t-test; ns not significant; n = 4 biological replicates. Source data are available online for this figure.

analysis (Dataset EV1), while well expressed in HIV-1 producing cells, confirming a low level of cellular contamination in the viral preparation. Western blot analysis further confirmed the enrichment of LC3B, GABARAPL1 and GABARAP in viral preparation produced from HeLa cells. GABARAPL2 was also found in HIV-1 preparation from HeLa cells, albeit to a lower extent compared to other LC3/GABARAP proteins, and was not associated with virions from CD4 + T cells, confirming the weak enrichment of this protein found in our LC-MS analysis (Fig. 2B). Of note, the presence of these ATG8 proteins in mock preparation validated previous reports showing that these proteins are associated with extracellular vesicles (Leidal et al, 2020).

The ATG8 (LC3/GABARAP) proteins are present in both non-lipidated cytoplasmic form (ATG8-I) (higher band) or in membrane-anchored lipidated form (ATG8-II) (lower band). The lipidated form resides in the membrane of the autophagosome (inner and outer), but also on the membrane of phagosomes, endosomes or secretory vesicles, during CASM processes (LAP and LANDO) and unconventional secretion (Galluzzi and Green, 2019). We noticed that non-lipidated (ATG8-I) and lipidated (ATG8-II) forms of ATG8 proteins can be associated with HIV-1 preparations, with different intensities depending on the cell type. The lipidated forms of LC3B (LC3B-II) and GABARAP (GABARAP-II and GABARAPL1-II) proteins were enriched in virion preparations from HeLa cells (Fig. 2C). However, non-lipidated forms of GABARAP (GABARAP-I), GABARAPL1 (GABARAPL1-I), and LC3B (LC3B-I) can also be detected with the corresponding lipidated forms in virions prepared from infected CD4 + T cells (Fig. 2D), suggesting that the two forms of ATG8 proteins could be associated with virions depending on the cell type producing virions.

To confirm a specific association of ATG8 proteins with HIV-1 viral particles, we set up an immunoprecipitation assay capturing HIV-1 virions produced from HeLa cells infected with VSVg-pseudotyped HIV-1, using an antibody against the HIV-1 surface envelope (Env) glycoprotein (SUgp120). We focused on LC3B and GABARAPL1, the most enriched ATG8 proteins in our viral preparation. SUgp120 immunoprecipitation (IP) of virions showed an enrichment of mature structural HIV-1 proteins, SUgp120 Env, and MAp17 Gag subproduct, and of LC3B and GABARAPL1 proteins, compared to the control IP with an IgG isotype (Fig. 2E). LC3B is associated with HIV-1 viral particles, mainly under its

lipidated form, and GABARAPL1 is present mainly under its non-lipidated form (Fig. 2E). This result confirmed that LC3B and GABARAPL1 proteins are incorporated and can be found associated under both their lipidated (LC3B and GABARAPL1) and non-lipidated (only GABARAPL1) form into HIV-1 virions.

To further distinguish whether these virion-associated factors reside within viral particles or are attached to their surface, isolated virions were subjected to subtilisin A treatment (Perez-Caballero et al, 2009). As expected, after treatment, the full-length 120 kDa SUgp120 exposed to the surface of viral particle was no longer detectable by immunoblotting, while the structural precursor Pr55Gag and these subproducts p41 and CAp24 protected inside the virions from the serine protease digestion, were detected (Fig. 2F). Moreover, the two forms of GABARAPL1 and LC3B associated with HIV-1 preparations were still detected after subtilisin treatment, indicating their incorporation inside the particles found in HIV-1 preparations.

It has been previously shown that the HIV-1 accessory proteins, Nef and Vif, are able to bind, respectively, with all GABARAP family members (Boeske et al, 2017) and with LC3B (Borel et al, 2015). We thus checked whether Nef, Vif or the other accessory proteins Vpu and Vpr modulate the incorporation of ATG8 proteins into the viral particle. Culture supernatants of Vpu (Udel)-, Vif (ΔVif)-, Vpr (ΔVpr)-, Nef (ΔNef)-deleted HIV-1 provirus transfected HeLa cells (Fig. EV2A) were ultracentrifuged on a 20% sucrose cushion. As expected, less virions are produced in the absence of Vpu, due to the absence of Vpu-countermeasure against BST2/Tetherin, a restriction factor trapping the virus at the cell surface (Neil et al, 2008). Western blot analysis of these viral preparations showed that the absence of accessory gene expression did not affect the incorporation of LC3 and GABARAP family members into viral particles, indicating that ATG8 proteins are incorporated inside HIV-1 particles independently of viral accessory proteins.

Altogether, amongst the ATG8 proteins, LC3B and GABARAPL1 were found significantly enriched in HIV-1 virions produced from HeLa cells and CD4 + T cells, one of the natural targets of HIV-1.

## GABARAP proteins are important for the production of infectious HIV-1 particles

To next investigate the role of ATG8 proteins in the morphogenesis of HIV-1 virions, we monitored HIV-1 production after a single round of infection of HeLa cells depleted for each member of the

LC3/GABARAP family by siRNA. Depleted cells were infected with VSVg-pseudotyped NL4-3 HIV-1. Forty-eight hours later, cells were harvested, and supernatants were collected and ultracentrifuged on a 20% sucrose cushion. Depletion of LC3/GABARAP family members was verified by RT-qPCR (Fig. EV3A–F) and western blot when specific antibodies were available (LC3B, GABARAP, GABARAPL1, and GABARAPL2) (Fig. 3A,B). We first noticed no detectable expression of LC3A mRNA in HeLa cells, whereas it is well expressed in T cells (LT) (Fig. EV3A). By contrast, the other LC3/GABARAP proteins were well expressed in HeLa cells and efficiently knocked down after siRNA treatment (Figs. 3A,B and EV3B–F). Released HIV-1 virions were then assessed by ELISA quantification of the viral CAp24 protein, which is derived from the structural Gag precursor, present in the producer cells and in the cell supernatants. Calculation of HIV-1 release index, corresponding to the ratio of released CAp24 to cell-associated CAp24, revealed that HIV-1 release was decreased following LC3C depletion (~50%), compared to cells treated with control siRNA (siCTRL), and not affected after depletion of other human LC3/GABARAP family members (Fig. 3C,D). This effect was the result of an accumulation of cell-associated HIV-1 CAp24 and a decrease of HIV-1 released in the supernatant (Fig. 3A, lane 8) and corroborates our previous studies showing that LC3C contributes to Vpu-mediated antagonism of BST2/Tetherin restriction on HIV-1 release (Madjo et al, 2016). We next scored the infectious titers of viral particles produced by cells depleted for each LC3/GABARAP protein (Fig. 3E,F) by infection of HeLa P4R5 indicator cells and normalized them to the amount of CAp24 present in the cell supernatants (referred herein as HIV-1 infectivity). While silencing of LC3 proteins and GABARAPL2 did not affect the infectivity of produced viral particles (Fig. 3E,F), a reproducible, albeit not statistically significant, decrease in HIV-1 infectivity was observed upon GABARAP and GABARAPL1 knockdown, compared to control cells (siCTRL), suggesting that GABARAP and GABARAPL1 may be involved in the production of infectious viral particles. The GABARAP proteins are highly similar in both sequences (over 75% within the GABARAP subfamily) and structures (Johansen and Lamark, 2020) (Fig. EV3G,H). This protein family is indispensable for the autophagic process, and each individual GABARAP protein is sufficient to drive autophagy (Nguyen et al, 2021), arguing that GABARAP family members share similar functions during autophagic processes. To avoid a functional redundancy of GABARAP subfamily members in our assay, we performed HIV-1 production assays in HeLa cell lines knockout for the three GABARAP proteins by the CRISPR-Cas9 system (GABARAP TKO, Fig. EV4A). GABARAP TKO HeLa cells were transfected with WT HIV-1 NL4-3 proviral DNA. As controls, we used parental HeLa cells and cells knockout for the three LC3 proteins (LC3 TKO; Fig. EV4A) (Nguyen et al, 2021). Knockout of LC3/GABARAP proteins and its impact on autophagy flux was checked by western blot (Fig. EV4A). Similarly to siRNA interference experiments targeting LC3C gene expression (Fig. 3C) and consistent with the contribution of LC3C in HIV-1 release in BST2/Tetherin expressing cells (Madjo et al, 2016), ELISA quantification of CAp24 in LC3 TKO cells yielded a decrease of HIV-1 release (~50 to ~60%), compared to the control (Fig. 4A–D). In contrast, a larger amount of viral particles was detected in the supernatant of HIV-1-producing GABARAP TKO cells that correlated with higher Gag expression in GABARAP TKO cells

(Fig. 4A–C). Hence, as shown using individual siRNA targeting each GABARAP subfamily member (Fig. 3D), the knockout of the three GABARAP subfamily members did not overall affect HIV-1 release (GABARAP TKO, Fig. 4D). We next determined the infectivity of LC3 and GABARAP TKO cell culture supernatants, as described above. Whereas knockout of LC3 genes slightly increased HIV-1 infectivity, we observed that the infectivity of viral particles produced by GABARAP TKO cells is decreased by 50%, compared to the supernatant from transfected parental cells (Fig. 4E), suggesting that GABARAP proteins play a role in the production of fully infectious HIV-1 particles.

To further characterize the impact of GABARAP subfamily members knockout on HIV-1 infectivity, we then analyzed the composition of viral particles produced by GABARAP TKO cells. Western blot analysis of equal volumes of viral preparations produced from GABARAP TKO cells showed an increase in the amount of CAp24 and SUgp120 Env proteins compared to the control preparation (Fig. 4A), consistent with the CAp24 increase detected by ELISA in viral suspension in Fig. 4C. Neither Env incorporation default nor Gag processing alteration were observed in this viral preparation (Fig. 4A), indicating that the low HIV-1 infectivity of viral particles produced form GABARAP subfamily members KO cells is not due to a default in Env incorporation or Gag maturation.

Both non-lipidated (GABARAPL1-I) and lipidated form (GABARAPL1-II) of GABARAPL1 are incorporated into HIV-1 particles (Fig. 2E). We thus investigated whether the role of GABARAP subfamily members in HIV-1 infectivity depends on the lipidation of these proteins. The lipidation of GABARAP proteins is mediated by specific ATG proteins, notably the activity of the ATG5-ATG12-ATG16L1 multimeric complex (Mizushima et al, 2011). We thus analyzed the infectivity of HIV-1 particles produced from ATG5−/− KO cells (Judith et al, 2023) in which LC3/GABARAP proteins are not lipidated (Fig. EV4B). HIV-1 NL4-3 provirus was transfected into parental and ATG5−/− KO HeLa cells, and HIV-1 infectivity was measured on ultracentrifuged HIV-1-containing supernatant. Although only the non-lipidated form of LC3B, GABARAP, and GABARAPL1 were found enriched in the supernatants from HIV-1 producing ATG5−/− KO cells (Fig. EV4B), the infectivity of corresponding viral preparations was unchanged or slightly increased compared to virus produced from parental cells (Fig. EV4C). Thus, the role of GABARAP proteins in the production of fully infectious HIV-1 particles does not depend on their lipidation.

Altogether, the data showed that GABARAP subfamily members are important for the production of infectious HIV-1 particles. This function of GABARAP proteins is independent of their lipidation state and their role in canonical autophagy.

## GABARAP family proteins modulate the packaging of viral gRNA

Contrary to LC3 subfamily proteins, the role of GABARAP proteins in the HIV-1 replication cycle, particularly in viral infectivity, has remained largely unexplored. We therefore sought to investigate in depth the GABARAP-associated phenotype affecting the infectivity of viral particles. As a first step, we examined whether the reduced infectivity of viral particles produced from GABARAP TKO cells (Fig. 4E) could be attributed to altered packaging of viral gRNA into budding virions. The amount of viral gRNA present in GABARAP TKO cells and in the corresponding viral preparations

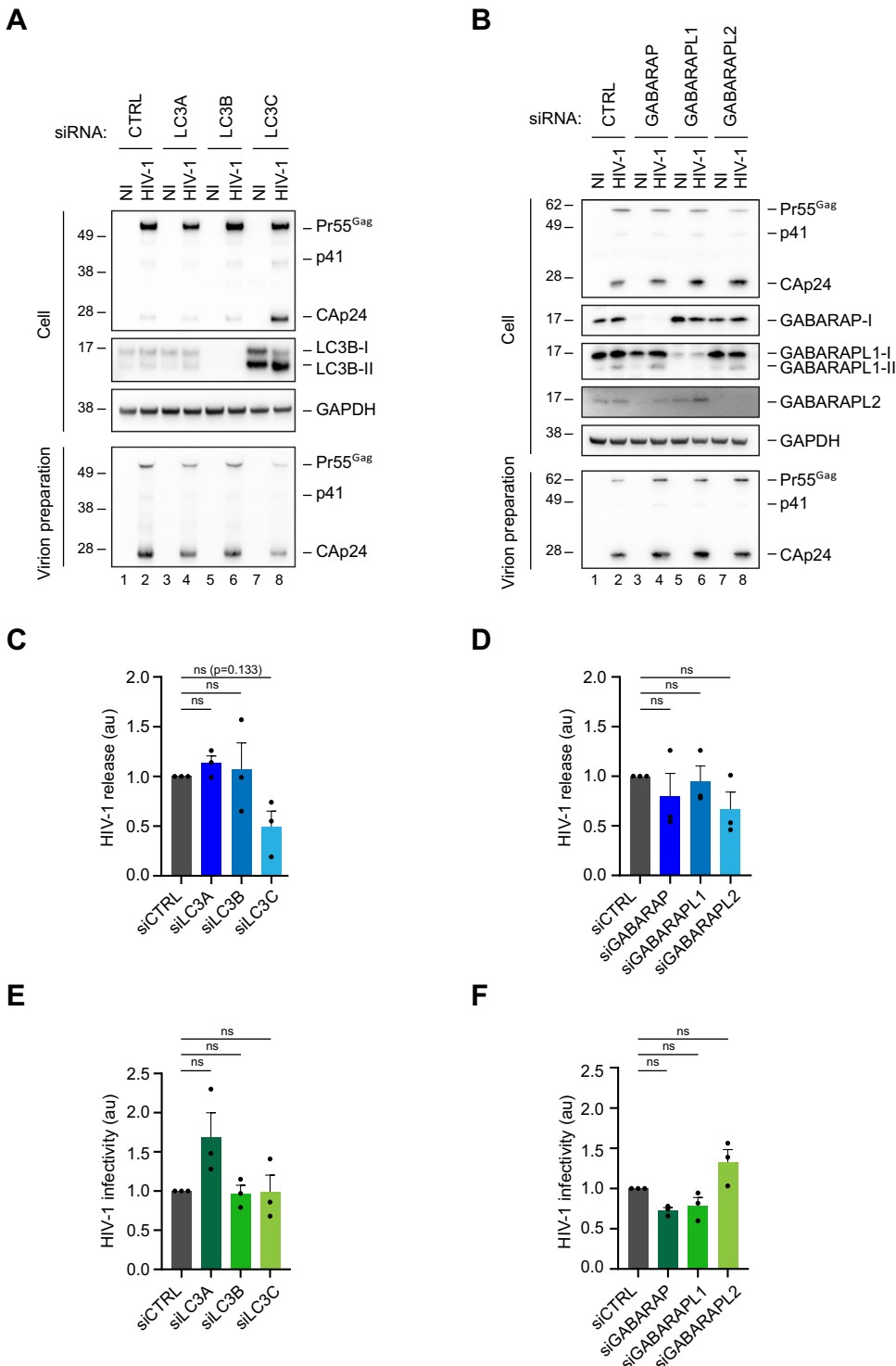

was measured by real-time PCR (Fig. 5A,B) and normalized to the amount of CAp24 released to evaluate viral gRNA packaging efficiency (Fig. 5C). We first noticed that, along with the increase of cell-associated CAp24 (Fig. 4B), GABARAP TKO cells exhibited a higher level of viral gRNA in comparison to parental cells (Fig. 5A). This may either suggest that GABARAP TKO cells are more permissive to HIV-1 provirus transfection or that GABARAP

family proteins depletion induces an accumulation of gRNA. Interestingly, we observed that knockout of GABARAP family proteins decreases viral gRNA packaging (~40%; Fig. 5C) into viral particles, but not those of 7SL RNA, a cellular RNA known to be enriched in HIV-1 virions (Eckwahl et al, 2016; Onafuwa-Nuga et al, 2006; Didierlaurent et al, 2011). Calculation of RNA release index, corresponding to the ratio of released RNA to cell-associated

**Figure 3.    Effect of LC3/GABARAP family member depletion on the production of infectious HIV-1 particles.**

(A) HeLa cells transfected with control siRNA (CTRL) or siRNA targeting LC3A, LC3B or LC3C were infected or not (NI) with a VSVg pseudotyped WT NL4-3 HIV-1 at a M.O.I. of 0.5. Cell lysates were recovered, and supernatants were collected 48 h later, filtered and ultracentrifuged on a 20% sucrose cushion. Western blot analysis of HIV-1 Gag and CAp24 products, LC3B and GAPDH in producing cells and supernatants. All western blots are representative of at least three biological replicates. In all western blot, LC3B-I, GABARAP-I, and GABARAPL1-I correspond to the non-lipidated form of these proteins and LC3B-II, GABARAP-II and GABARAPL1-II to the lipidated forms. (B) HeLa cells transfected with control siRNA (CTRL) or siRNA targeting GABARAP, GABARAPL1 or GABARAPL2 were infected with a VSVg pseudotyped WT NL4-3 HIV-1 at an MOI of 0.5. Cell lysates were recovered, and supernatants were collected 48 h later, filtered and ultracentrifuged on a 20% sucrose cushion. Western blot analysis of HIV-1 Gag and CAp24 products, GABARAP, GABARAPL1, GABARAPL2, and GAPDH in producing cells and supernatants. All western blots are representative of at least three biological replicates. HIV-1 release of viral production described in A and in B was calculated as the ratio between released CAp24 and cell-associated CAp24 quantified by ELISA, in (C, D), respectively. Results were normalized to control cells. The infectivity of the released virus in productions described in (A, B) was determined by a β-galactosidase reporter assay and normalized by the quantity of released CAp24, in (E,F), respectively. Data information: In (C–F), statistical analysis was performed using a one-way ANOVA with Dunnett's multiple comparisons test; mean ± SEM; n = 3 biological replicates; ns not significant. Source data are available online for this figure.

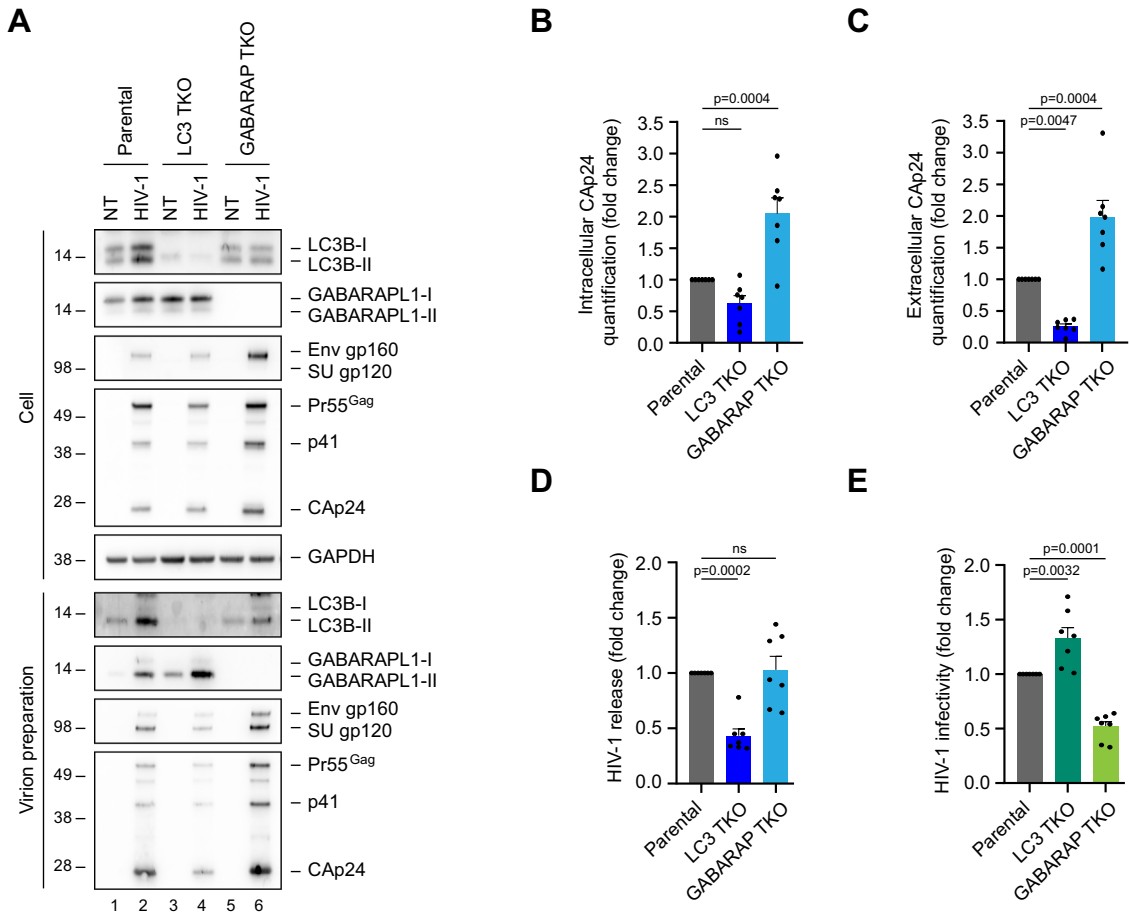

**Figure 4.    GABARAP proteins are required for the production of infectious HIV-1 particles.**

Parental, LC3 (LC3 TKO), or GABARAP (GABARAP TKO) knockout cells were transfected or not (NT) with WT NL4-3 HIV-1 proviral DNA. Cell lysates were recovered, and supernatants were collected 48 h later, filtered and ultracentrifuged on a 20% sucrose cushion. (A) Western blot analysis of LC3B, GABARAPL1, HIV-1 SUgp120, HIV-1 Gag, and CAp24 products and GAPDH in producing cells and supernatants. All western blots are representative of at least three biological replicates. LC3B-I and GABARAPL1-I correspond to the non-lipidated form of LC3B and GABARAPL1, and LC3B-II and GABARAPL1-II to the lipidated forms. Intracellular (B) and extracellular (C) CAp24 were quantified by ELISA. Results were normalized to the control condition. (D) HIV-1 release was calculated as the ratio between released CAp24 and cell-associated CAp24 quantified by ELISA. Results were normalized to the control condition. (E) The infectivity of the released virus was determined by a β-galactosidase reporter assay and normalized by the quantity of released CAp24. Results were normalized to the control condition. Data information: statistical analysis was performed using a one-way ANOVA with Dunnett's multiple comparisons test; mean ± SEM; n = 7 biological replicates. ns not significant. Source data are available online for this figure.

RNA, confirmed the decrease (~50%) of gRNA into HIV-1 containing supernatant (Fig. 5D).

We further examined the gRNA packaging in purified HIV-1 viral particles. Parental and GABARAP TKO cells were infected

with VSVg-pseudotyped HIV-1 virions, and the amount of viral gRNA present in cells and in Env-immunoprecipitated HIV-1 viral particles was analyzed by real-time PCR, as described in Fig. 2E. As shown in Fig. 5E, the virion-packaged gRNA level was decreased

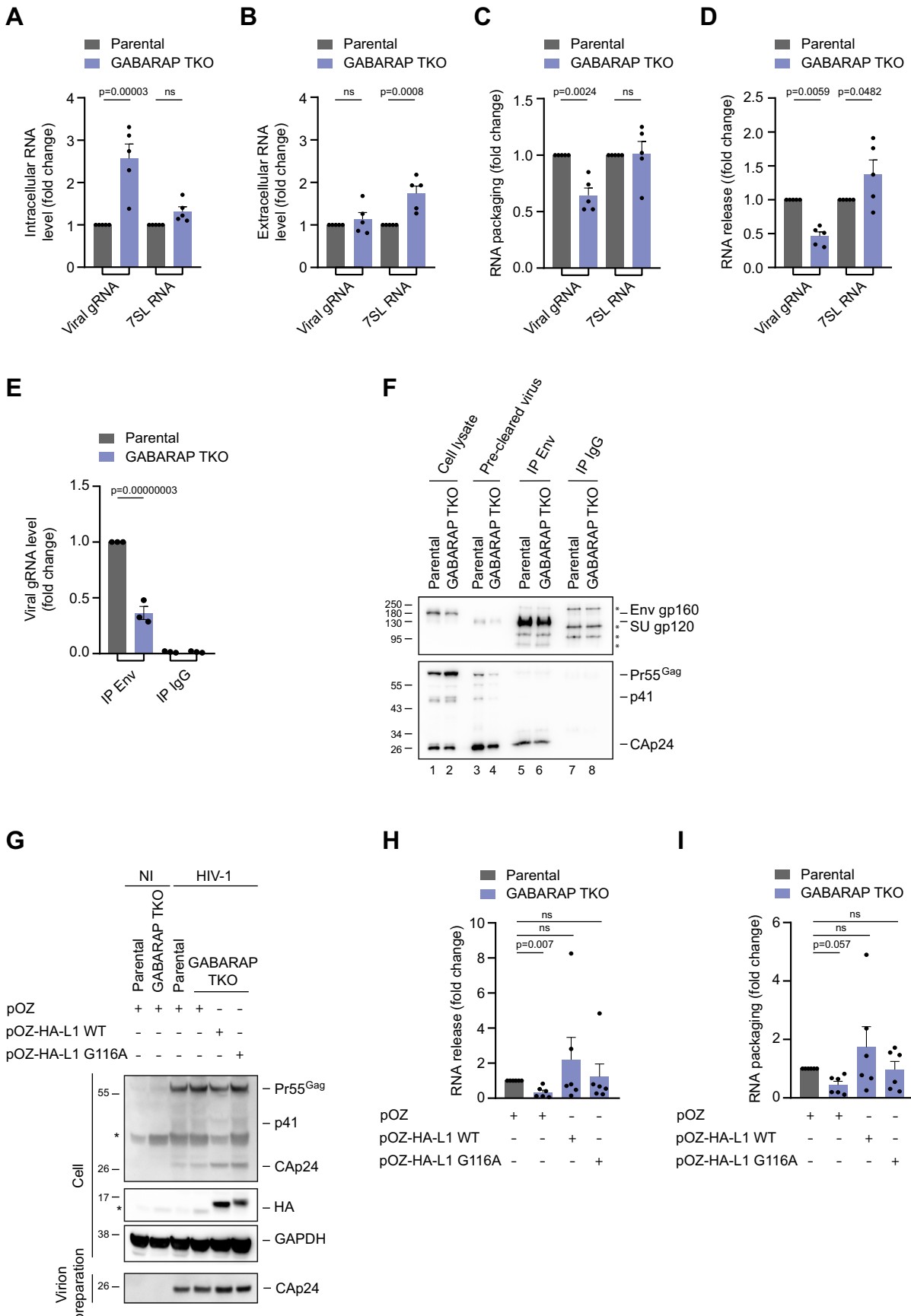

**Figure 5. GABARAP proteins are involved in the packaging of HIV-1 gRNA.**

Parental or GABARAP (GABARAP TKO) knockout cells were transfected with WT NL4-3 HIV-1 proviral DNA. Cell lysates were recovered, and supernatants were collected 48 h later and ultracentrifuged on a 20% sucrose cushion. Intracellular (**A**) and extracellular (**B**) viral gRNA and 7SL RNA were determined by RT-qPCR of total RNA extracted from cells and supernatants. RNAs were normalized to the housekeeping product actin mRNA for cellular samples, and to mouse actin mRNA for supernatant samples. Results were normalized to the control condition. (**C**) The quantity of viral gRNA and 7SL RNA packaged into viral particles was calculated as the ratio between the quantity of released viral gRNA and the quantity of released CAp24. (**D**) Release of viral gRNA and 7SL RNA was calculated as the ratio between released RNA and cell-associated RNA. Data information: In (**B–D**), statistical analysis was performed using a one-way ANOVA with Sidak's multiple comparison test; mean ± SEM; $n = 5$ biological replicates; ns not significant. (**E**) HeLa cells were infected with VSVg pseudotyped NL4-3 HIV-1 at an MOI of 1. Supernatants were collected 48 h later and ultracentrifuged. IP of HIV-1 particles using human monoclonal anti-SUgp120 HIV-1 or human IgG1k as control. Viral gRNA from the IP was quantified by RT-qPCR and normalized to precleared virus fractions and control IP. Statistical analysis was performed using a one-way ANOVA with Sidak's multiple comparisons test; mean ± SEM; $n = 3$ biological replicates. (**F**) Western blot analysis of HIV-1 SUgp120 and HIV-1 Gag and CAp24 products, of cell lysates, precleared virus and IPs. All western blots are representative of at least two biological replicates. Asterisks represent nonspecific bands. (**G**) HeLa GABARAP (GABARAP TKO) knockout cells were transduced with empty retroviral vector (pOZ) or retroviral vectors encoding either HA-tagged wild-type (pOZ-HA-L1 WT) or non-lipidated mutant form (pOZ-HA-L1 G116A) of GABARAPL1 and selected with puromycin to establish stable cell lines. Parental, GABARAP TKO and HA-GABARAPL1 WT or G116A trans-complemented TKO stable cell lines were then infected with VSV-G pseudotyped NL4-3 HIV-1 for 2 h at a multiplicity of infection (MOI) of 0.3. Cell lysates were recovered, and supernatants were collected 48 h later and ultracentrifuged on a 20% sucrose cushion. Western blot analysis of HA, HIV-1 Gag, and CAp24 products and GAPDH in producing cells and supernatants. All western blots are representative of at least six biological replicates. Asterisks (*) represent nonspecific bands. (**H**) Release of viral gRNA was calculated as the ratio between released RNA and cell-associated RNA. Released and cell-associated viral gRNA were determined by RT-qPCR of total RNA extracted from cells and supernatants. RNAs were normalized to the housekeeping product actin mRNA for cellular samples, and to mouse actin mRNA for supernatant samples. Results were normalized to the control condition. (**I**) The quantity of viral gRNA packaged into viral particles was calculated as the ratio between the quantity of released viral gRNA and the quantity of released CAp24 quantified by ELISA. Data information: In (**H, I**), statistical analysis was performed using the Kruskal–Wallis test followed by Dunn's multiple comparison test; mean ± SEM; $n = 6$ biological replicates; ns not significant. Source data are available online for this figure.

( ~ 60%) in the absence of GABARAP family protein expression, whereas similar amounts of viral particles were captured with SUgp120 antibodies from supernatants of HIV-1-infected parental and GABARAP TKO cells (Fig. 5F, lanes 5 and 6).

Finally, to exclude potential off-target effects or adaptive changes in GABARAP TKO cells, we performed trans-complementation assays using retroviral transduction to re-express either wild-type or non-lipidated mutant form of GABARAPL1 proteins in the TKO background. Stable cell lines expressing HA-tagged wild-type GABARAPL1 (pOZ-HA-L1 WT), its corresponding C-terminal glycine-to-alanine mutant (which prevents lipidation; pOZ-HA-L1 G116A), or the corresponding empty vector (pOZ), were generated and validated (Fig. EV5). We then assessed HIV-1 gRNA release and packaging efficiency 48 h post-infection of these trans-complemented cell lines. As expected, GABARAP TKO cells showed a marked reduction in both gRNA release and packaging. Reexpression of wild-type GABARAPL1 protein restored both phenotypes, confirming its involvement in HIV-1 gRNA packaging (Fig. 5G–I). Importantly, expression of the non-lipidated GABARAPL1 (G116A) mutant also partially rescued HIV-1 gRNA packaging and release to a comparable extent as the wild-type protein, indicating that the role of GABARAPs in this process is independent of their lipidation (Fig. 5G–I).

These findings strongly support our conclusion that GABARAP proteins are directly involved in the packaging of HIV-1 gRNA, and that this novel function is independent of their membrane conjugation.

## GABARAP proteins interact with HIV-1 Gag and gRNA

Packaging of the HIV-1 genome into a viral particle requires interactions between the dimeric 5'UTR of gRNA and the assembling Gag polyproteins (Sumner and Ono, 2024; Gorelick et al, 1993; Wu et al, 2013; Paillart et al, 1996). Our observations of the effect of GABARAP protein knockout on HIV-1 gRNA packaging led us to examine whether GABARAP proteins could specifically interact with HIV-1 Gag or gRNA, or modify Gag:gRNA interaction. We found through co-IP experiments that all GABARAP proteins interact efficiently with Pr55Gag and its

subproducts p41 and CAp24 (Fig. 6A). Interestingly, RNase treatment showed that the interaction between Gag and GABARAPL1, the most abundant GABARAP proteins found in HIV-1 particles, strongly depends on the presence of RNAs (Fig. 6B,C, lanes 11 vs 12), whereas Gag:LC3B interaction does not (lanes 15 vs 16). This latter result led us to explore whether GABARAPL1 interacts directly with HIV-1 gRNA. Using ultra-violet light cross-linking of protein and RNA that are directly interacting in culture cells, followed by RNA immunoprecipitation (UV-RIP) assay, we observed that endogenous GABARAPL1 binds directly to HIV-1 gRNA in VSVg-pseudotyped HIV-1 infected parental HeLa cells (Fig. 6D,E), suggesting that GABARAPL1: HIV-1 gRNA interaction could favor the association of GABARAPL1 to Gag.

## GABARAP proteins control the fate of HIV-1 gRNA

Considering the interaction of GABARAPL1 with HIV-1 gRNA, we next examined whether GABARAP proteins could impact the Gag:gRNA interaction, essential for HIV-1 gRNA packaging. UV-RIP experiments were performed with an antibody directed against Pr55Gag/CAp24, in VSVg-pseudotyped HIV-1 infected parental and GABARAP TKO cells. We found that knocking out GABARAP proteins did not decrease the binding of Pr55Gag to HIV-1 gRNA (Fig. 6F,G). On the contrary, the absence of GABARAP proteins leads to the accumulation of Gag:gRNA complexes in HIV-1-infected cells. Finally, we explored whether GABARAP proteins influence the subcellular distribution HIV-1 gRNA in HIV-1 infected cells, notably the association of gRNA at the membranes. HIV-1-infected control and GABARAP TKO cells were fractionated, and cytosolic and membrane fractions were recovered (Fig. 7A,B). GAPDH and Transferrin receptor (TF-R) were used as controls for the cytosolic and membrane fractions, respectively. The amount of viral gRNA present in these fractions was measured by real-time PCR and normalized to the amount of 7SL RNA to evaluate the subcellular distribution of viral gRNA in the different fractions. Interestingly, we found that the association of HIV-1 gRNA with the membranes is markedly decreased in cells that do

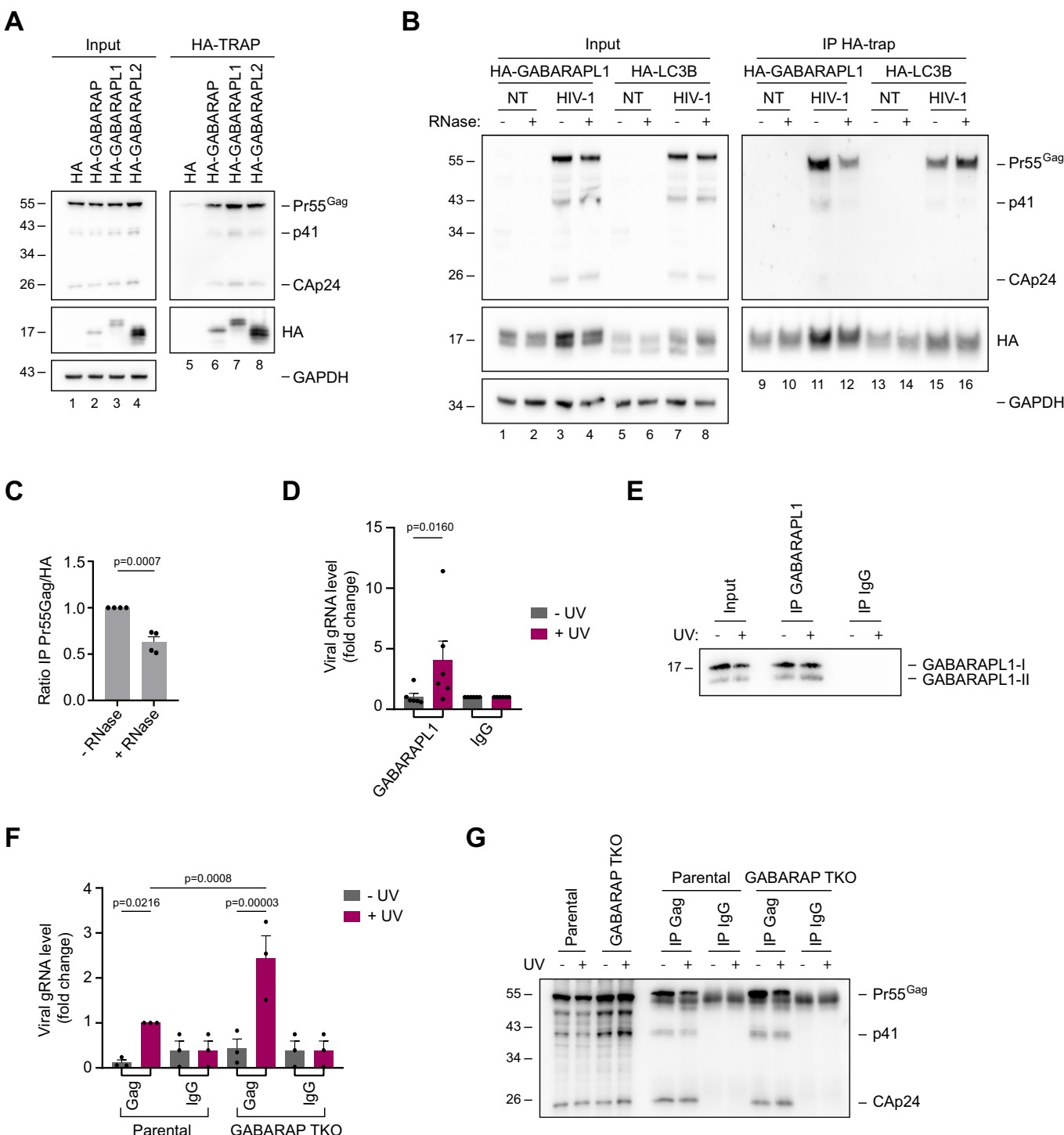

not express GABARAP proteins, revealing a contribution of GABARAP proteins in the targeting of HIV-1 gRNA at cellular membranes (Fig. 7A,B). To complement these results, we conducted RNAscope experiments in both parental and GABARAP TKO cells infected with VSV-G-pseudotyped HIV-1 WT (MA-YFP) to localize HIV-1 gRNA and structural components. Despite a global accumulation of HIV-1 gRNA in GABARAP TKO cells (Fig. 7C,D), HIV-1 gRNA puncta were located, significantly, farther from the plasma membrane than in parental cells (Fig. 7E),

suggesting an inefficient recruitment of HIV-1 gRNA to the viral assembly sites at the plasma membrane.

Overall, our data indicate that GABARAPL1 is associated with HIV-1 gRNA. In the absence of GABARAP proteins, Gag:gRNA complexes accumulate intracellularly and HIV-1 gRNA is less enriched in membrane fractions, spatially separated from the plasma membrane, suggesting that GABARAP proteins may facilitate the targeting of these complexes to assembly sites and promote efficient gRNA packaging into viral particles.

**Figure 6. GABARAP proteins interact with HIV-1 Gag and gRNA.**

(A) HA-trap Immunoprecipitation (IP) in HeLa cells extract co-transfected with plasmids expressing WT NL4-3 HIV-1 proviral DNA and pCMV-HA, HA-GABARAP, -GABARAPL1, or -GABARAPL2. Immunoprecipitated proteins were detected by western blotting of HIV-1 Gag and CAp24 products, HA and GAPDH. The higher band of HA-tagged GABARAP proteins corresponds to their non-lipidated form, and the lower corresponds to their lipidated form. All western blots are representative of at least three biological replicates. (B) HA-trap IP in HeLa cells, treated or not with RNase, extracts co-transfected with plasmids expressing WT NL4-3 HIV-1 proviral DNA and HA-GABARAPL1 or HA-LC3B. Immunoprecipitated proteins were detected by western blotting of HIV-1 Gag and CAp24 products, HA and GAPDH. All western blots are representative of at least three biological replicates. (C) Quantification of the ratio between Pr55Gag and HA protein levels in (B). Statistical analysis using an unpaired Student's $t$-test; mean ± SEM; $n = 4$ biological replicates. (D) HeLa parental cells were infected with VSVg-pseudotyped HIV-1 viruses at a MOI of 1. Forty-eight hours later, cells were UV-crosslinked. RNA was immunoprecipitated with antibodies against GABARAPL1 or IgG control. Viral gRNA from RIP were quantified by RT-qPCR and normalized to inputs and control immunoprecipitation. Statistical analysis using a one-way ANOVA with Sidak's multiple comparisons test; mean ± SEM; $n = 6$ biological replicates. (E) Western blot analysis from RIP in (D) of GABARAPL1. GABARAPL1-I corresponds to the non-lipidated form of GABARAPL1 and GABARAPL1-II to the lipidated form. All western blots are representative of at least three biological replicates. (F) Parental HeLa and GABARAP TKO cells were infected with VSVg-pseudotyped HIV-1 viruses at an MOI of 1. Forty-eight hours later, cells were UV-crosslinked. RNA was immunoprecipitated with antibodies against Gag or serum control. Viral gRNA from RIP were quantified by RT-qPCR and normalized to inputs and control immunoprecipitation. Statistical analysis using a one-way ANOVA with Sidak's multiple comparisons test; mean ± SEM; $n =$ three biological replicates. (G) Western blot analysis from RIP in (F) of HIV-1 Gag and CAp24 products. All western blots are representative of at least three biological replicates. Source data are available online for this figure.

# Discussion

Recent advances have revealed that some autophagy-related proteins (ATG), beyond their canonical role in bulk cytosolic degradation, also participate in diverse cellular processes such as CASM, unconventional protein secretion, extracellular vesicle release, antimicrobial responses, and viral replication and egress (Galluzzi and Green, 2019; Durgan and Florey, 2022). Several viruses hijack ATG-mediated exocytosis pathways to exit infected cells or use ATG-decorated membranes for their envelopment (Münz, 2021). Components of the autophagy machinery, including ATG8 family members and core ATG proteins such as ATG5, ATG7, and ATG16L1, have been implicated in replication cycles of both RNA and DNA viruses, facilitating viral dissemination and immune evasion (Abernathy et al, 2019; Beale et al, 2014; Reggiori et al, 2010). In contrast, the role of ATG proteins in HIV-1 egress remains underexplored.

Here, we employed the data-independent acquisition (DIA) mode of LC-MS analysis on purified HIV-1 preparations to identify host proteins potentially involved in viral assembly and release. Among the 2919 cellular factors identified, 1838 proteins were enriched with a fold change (FC) ≥1.3 and 46 proteins uniquely present in HIV-1 preparations (Figs. 1 and EV2A). Our approach confirmed the presence of known HIV-1-associated proteins (von Schwedler et al, 2003), including the ESCRT-I (TSG101, VPS28, and MVB12 A), ESCRT-III (CHMP4A and CHMP4B), and ESCRT-associated PDCD6IP/Alix proteins, validating our approach. With 1140 significatively enriched proteins associated with HIV-1 preparations being newly identified (Fig. 1E), our mass spectrometry approach significantly expands the HIV-1 proteome database.

Gene Ontology term enrichment pointed to processes including translation, protein folding, catabolism, intracellular trafficking, cytoskeleton organization, and macroautophagy (Fig. 1D). Among proteins linked to autophagy, we found 18 proteins associated with the ESCRT pathway and 23 with the autophagy machinery (Fig. 2A,B), notably, ATG9A involved in nucleation, ULK1 and ATG101 involved in the autophagy initiation, ATG3, ATG7, and ATG16L1 implicated in the elongation step of autophagy. ULK1, ATG101, and ATG3 have not previously been implicated in the HIV-1 viral cycle, others, like ATG9A and ATG7, has been shown to promote HIV-1 infectivity (Mailler et al, 2019) or dissemination as proviral factors (Brass et al, 2008), pointing toward a role in late replication stages that warrants further investigation.

Interestingly, we identified four of the six ATG8 family proteins (LC3B, GABARAPL1, GABARAP, and GABARAPL2) in HIV-1 preparations (Fig. 2A,B), with LC3B and GABARAPL1 proteins reaching enrichment levels comparable to key ESCRT proteins (Fig. 2B). Western blot and immunocapture experiments confirmed the presence of LC3B, GABARAPL1, and GABARAP in virions produced from HeLa cells and CD4 + T cells (Fig. 2C–E). This enrichment, previously undetected in other proteomic analyses of HIV-1 virions (Brégnard et al, 2013; Chertova et al, 2006; Gale et al, 2019; Ott et al, 2000; Santos et al, 2012; Saphire et al, 2006), may result from the implementation of DIA mode in our analysis, which allows the detection of low-abundance peptides from small or weakly represented proteins in sample. A recent report indicates that ATG8 proteins, particularly LC3B, are associated with exosomes (Leidal et al, 2020). We thus cannot entirely exclude the possibility that the presence of the LC3/GABARAP proteins in the HIV-1 proteome could result from the presence of exosomes in our HIV-1 preparation. However, the significant enrichment of LC3 and GABARAP proteins in subtilisin-treated HIV-1 particles, and their specific capture via Env-immunoprecipitation of HIV-1 virions, argue in favor of genuine incorporation into HIV-1 particles (Fig. 2E,F).

ATG8 proteins are present in cells either as a non-lipidated (cytosolic, ATG8-I) or lipidated (membrane-anchored, ATG8-II) forms. Both forms were detected in our viral preparations, with patterns varying depending on the cell type (Fig. 2C,D). Importantly, GABARAPL1 and LC3B were still detectable in HIV-1 preparations after subtilisin treatment, indicating their incorporation into viral particles (Fig. 2F). Notably, their incorporation appears independent of the viral Nef, Vif, Vpu, and Vpr accessory proteins (Fig. EV2), suggesting alternative mechanisms of recruitment. While LC3B has been shown to colocalize with HIV Gag-derived proteins (Kyei et al, 2009), our data now show that GABARAP proteins also interact with Gag and, in the case of GABARAPL1, with viral gRNA. These interactions are partly RNA-dependent (Fig. 6A–C) and likely mediate incorporation of GABARAP proteins into virions. The association of LC3/GABARAP proteins with HIV-1 particles adds to previous descriptions of LC3 protein association with the envelopes of dsRNA viruses, including EBV, HCMV, and VZV, as well as with vesicles secreted from cells infected with non-enveloped RNA viruses such as poliovirus (PV) and coxsackievirus B3 (CVB3), and

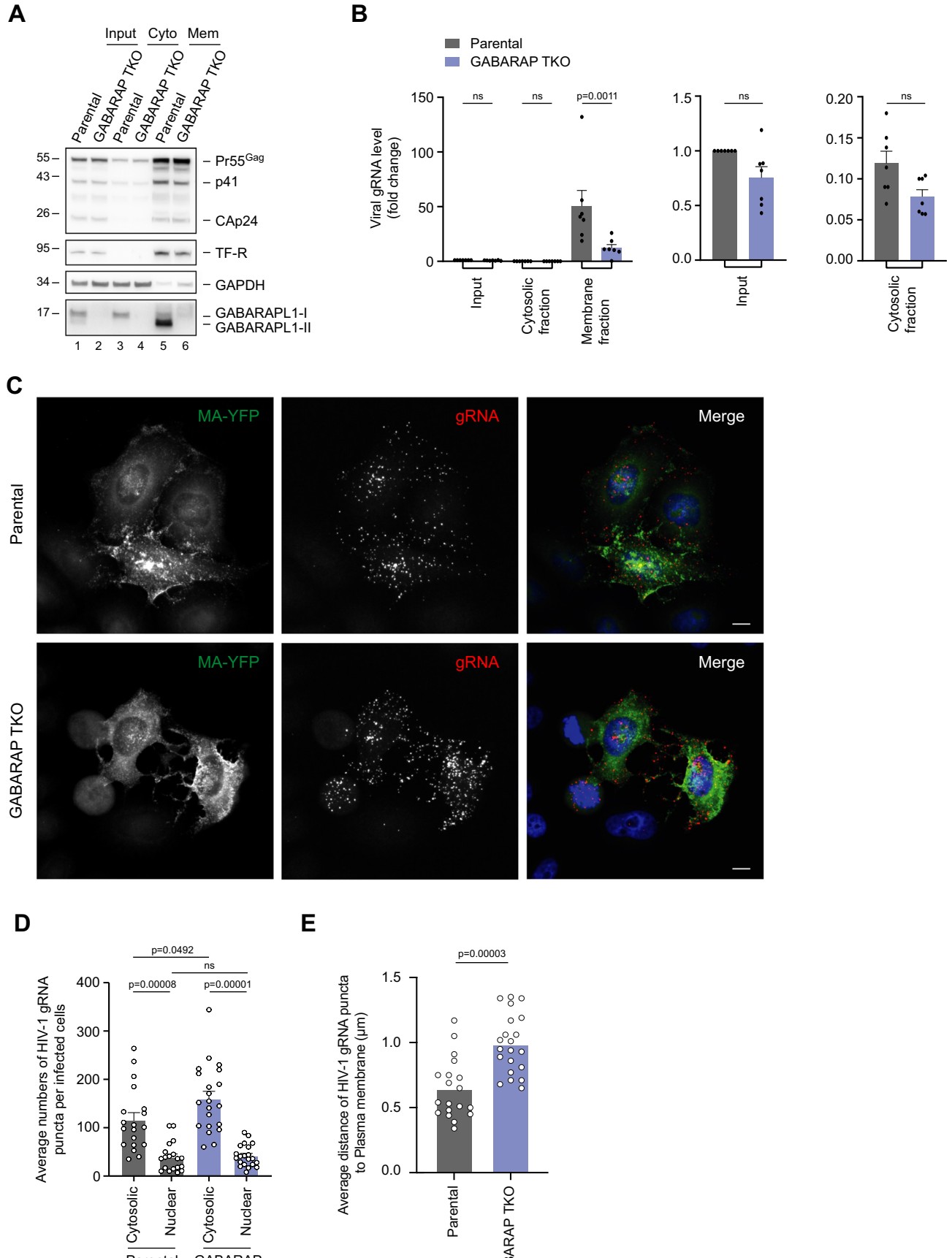

**Figure 7. GABARAP proteins contribute to the targeting of HIV-1 gRNA to cellular membranes in infected cells.**

(A) HeLa parental HeLa and GABARAP TKO cells were infected with VSVg pseudotyped HIV-1 viruses at an MOI of 1. Forty-eight hours later, cytosolic and membranes fractions were analyzed by immunoblotting with antibodies against GABARAPL1, Gag and CAp24 products, transferrin receptor (TF-R), GAPDH. GABARAPL1-I corresponds to the non-lipidated form of GABARAPL1 and GABARAPL1-II to the lipidated form. A representative image of three biological replicates is shown. Cyto cytosolic fraction, Mem membrane fraction. (B) Viral gRNA from the different fractions obtained in (A) were quantified by RT-qPCR and normalized to 7SL RNA. Statistical analysis using a one-way ANOVA with Sidak's multiple comparisons test; mean ± SEM; $n = 7$ biological replicates; ns not significant. (C) Confocal fluorescence microscopy of parental or GABARAP (GABARAP TKO) knockout cells infected with VSV-G-pseudotyped HIV-1 WT MA-YFP at a MOI of 0.5 for 24 h. Cells were then fixed and stained for HIV-1 gRNA using RNAscope reagents and DAPI (Scale bars 10 μm). Immunofluorescence images presented correspond to the mean intensity projection images generated by merging all the optical sections obtained from the z-stack using Fiji software. Images are representative of at least two experiments. gRNA puncta were quantified (D) and localized (E) across the full cell volume in 19 Parental and 21 GABARAP TKO infected cells using IMARIS software. Statistical analysis using one-way ANOVA with Sidak's multiple comparisons test (D) and an unpaired Student t-test (E); mean ± SEM. Source data are available online for this figure.

Flaviviridae viruses like dengue virus (DENV), hepatitis C virus (HCV), and Zika virus (ZIKV) (Abernathy et al, 2019).

Our functional assays revealed a specific and previously unreported requirement for GABARAP proteins in HIV-1 infectivity. Triple knockdown of GABARAP family members reduced infectivity by ~50% (Fig. 4E), whereas single knockdowns had milder effects (Fig. 3F), consistent with functional redundancy within the GABARAP subfamily, as also described in autophagy (Nguyen et al, 2021). Importantly, ATG5 knockout had no effect on infectivity (Fig. EV4C), indicating that this function does not rely on ATG8 lipidation or canonical autophagy (Mizushima et al, 2011). This aligns with recent studies in other viruses, including Coronaviruses (e.g., murine hepatitis virus, SARS-CoV) (Reggiori et al, 2010), and Arteriviruses (Monastyrska et al, 2013), suggesting a common viral strategy to exploit the non-canonical functions of ATG8 proteins.

We further demonstrated that the defect in infectivity of HIV-1 produced from GABARAP TKO cells results from impaired viral gRNA packaging into budding virions (Fig. 5C), but not from altered Env incorporation or Gag maturation (Fig. 4A). This defect in gRNA packaging was confirmed by immunocapture assays (Fig. 5E,F), and trans-complementation experiments showed that GABARAP proteins support this process independently of their lipidation status (Fig. 5G–I). Interestingly, this role in gRNA packaging is reminiscent of the emerging function of ATG8 proteins in the sorting of RNA-binding proteins (RBPs) and non-coding RNAs into extracellular vesicles (Leidal et al, 2020). gRNA packaging is a tightly regulated process involving Gag interactions with structured RNA motifs in the 5′ UTR of the viral genome and is required for the formation of infectious virions (Bernacchi, 2022; Sumner and Ono, 2024; Yasin et al, 2024; Didierlaurent et al, 2011; Nikolaitchik et al, 2021; Smyth et al, 2018). Besides serving as the viral genome, unspliced HIV-1 RNA is translated to produce Gag and Gag-Pol precursors, which orchestrate viral assembly. We observed intracellular accumulation of Gag precursors in TKO cells (Fig. 4A,B), suggesting that unspliced gRNA might be rerouted or misregulated in the absence of GABARAPs. Several studies have reported that differential transcriptional start site usage yields 1 G and 3Gs viral RNAs, which differ in their propensity for packaging versus translation (Brown et al, 2020; Ding et al, 2021; Duchon & Hu, 2024; Kharytonchyk et al, 2016; Nikolaitchik et al, 2023; Yasin et al, 2024). Previous studies showed that LC3/GABARAP proteins could regulate RNA metabolism by targeting certain RNAs for degradation (Hwang et al, 2022). Thus, GABARAP proteins might influence the balance between these gRNA species (1G or 3Gs). In this scenario, the absence of GABARAP proteins could alter the

gRNA species ratio in infected cells, changing the level of gRNA available for translation and disturbing the assembly process. This hypothesis is supported by the detection of increased Gag:gRNA complexes in TKO cells and reduced membrane-associated gRNA, suggesting a trafficking or targeting defect.

Selective packaging of the HIV-1 genome during virion assembly is facilitated by interactions between the dimeric 5′-end gRNA and the NC domain of a limited number of assembling viral Gag polyproteins (Abd El-Wahab et al, 2014; Bernacchi, 2022; Gorelick et al, 1993; Sumner and Ono, 2024; Wu et al, 2013). Moreover, correct assembly of the nascent immature Gag lattice is essential for selecting the gRNA and initiating its packaging (Lei et al, 2023). Here, we found that GABARAP protein knockout does not weaken Gag-gRNA binding per se (Fig. 6F,G); rather, it appears to impair the membrane localization of this complex (Fig. 7A–E). GABARAPL1 directly binds to viral gRNA (Fig. 6D,E) and interacts with Gag, partly in an RNA-dependent manner (Fig. 6A–C), which supports the hypothesis that it acts as a molecular adapter guiding the Gag:gRNA complex to assembly sites. GABARAP proteins are known to regulate the anterograde trafficking of receptors to the plasma membrane (Ye et al, 2021) and are associated with numerous RNA granule-associated proteins (Behrends et al, 2010). Previous studies have reported different Gag assembly intermediates associated with RNA granules containing cellular assembly facilitators, ABCE1 and DDX6, in the presence or absence of gRNA (Barajas et al, 2018; Reed et al, 2018). We therefore propose that GABARAP proteins facilitate the trafficking of Gag-gRNA complexes to the plasma membrane, and that their absence causes the accumulation of Gag precursor and viral gRNA in infected cells (Figs. 5A and 6F,G), their mislocalization (Fig. 7), reduced packaging (Fig. 5E,F) and defective virion infectivity (Fig. 4E). This phenotype is not observed with LC3 proteins, likely due to functional specializations within the ATG8 family. While LC3 proteins primarily regulate autophagosome elongation and autophagic cargo degradation, GABARAPs are more directly involved in membrane trafficking and receptor transport toward the plasma membrane. GABARAPs also display higher affinity for certain LIR-containing adapter proteins involved in endomembrane dynamics (Johansen and Lamark, 2020; Mansuy et al, 2004; Wang and Olsen, 2000), which may explain their unique ability to support gRNA targeting to budding sites.

Overall, we uncover a novel, non-canonical role for GABARAP proteins in the HIV-1 replication cycle, specifically in promoting gRNA packaging. This function is unique to GABARAP subfamily members and not shared by LC3 proteins, despite their structural similarities. Our findings expand the repertoire of ATG8 protein

functions in virus-host interactions and underscore the merging links between autophagy proteins, RNA regulation, and viral assembly.

# Methods

### Reagents and tools table

| Reagent/resource | Reference or source | Identifier or catalog number |
|---|---|---|
| **Experimental models** | | |
| HeLa cells (*H. sapiens*) | ATCC | CRM-CCL-2 |
| HEK293T cells (*H. sapiens*) | ATCC | CRL-3216 |
| HeLa P4R5 cells (*H. sapiens*) | NIH AIDS Research and Reference Reagent Program | ARP-3580 |
| HeLa Parental cells (*H. sapiens*) | Nguyen et al, J Cell Biol 2016 - PMID: 27864321 | N/A |
| HeLa LC3 TKO cells (*H. sapiens*) | Nguyen et al, J Cell Biol 2016 - PMID: 27864321 | N/A |
| HeLa GABARAP TKO cells (*H. sapiens*) | Nguyen et al, J Cell Biol 2016 - PMID: 27864321 | N/A |
| HeLa ATG5-/- 92#3 cells (*H. sapiens*) | Judith et al, PNAS 2023 - PMID: 37155854 | N/A |
| Primary CD4 + T cells (*H. sapiens*) | HIV-seronegative human buffy coats from Etablissement Français du Sang | N/A |
| **Recombinant DNA** | | |
| pCMV-HA | Gallois-Montbrun et al, J. Virol 2007 - PMID: 17166910 | N/A |
| pCMV-HA-LC3B | This study | N/A |
| pCMV-HA-GABARAP | This study | N/A |
| pCMV-HA-GABARAPL1 | This study | N/A |
| pCMV-HA-GABARAPL2 | This study | N/A |
| HIV-1 proviral DNA pNL4-3 | NIH AIDS Research and Reference Reagent Program | ARP-114 |
| HIV-1 proviral DNA pNL4-3 Udel | NIH AIDS Research and Reference Reagent Program | ARP-968 |
| HIV-1 proviral DNA pNL4-3 ΔVif | Karczewski et al J. Virol 1996 - PMID: 8523563 | N/A |
| HIV-1 proviral DNA pNL4-3 ΔVpr | Eckstein et al J Exp Med 2001 - PMID: 11714748 | N/A |
| HIV-1 proviral DNA pNL4-3 ΔNef | NIH AIDS Research and Reference Reagent Program | ARP-12755 |
| HIV-1 proviral DNA pNL4-3 (MA/YFP) | Jouvenet et al PLos Biol 2006 - PMID: 17147474 | N/A |
| VSVg expression vector (pMD.G) | Addgene | 187440 |
| pCMV-Gag-Pol vector | MyBioSource | MBS168500 |
| pOZ-FH-N | Sobhian et al Mol Cell 2010 - PMID: 20471949 | N/A |
| pOZ-FH-N-GABARAPL1 | This study | N/A |

| Reagent/resource | Reference or source | Identifier or catalog number |
|---|---|---|
| pOZ-FH-N-GABARAPL1 G116A | This study | N/A |
| **Antibodies** | | |
| Polyclonal swine anti-rabbit Immunoglobulins/HRP | Dako | P021702-2 |
| Polyclonal rabbit anti-mouse Immunoglobulins/HRP | Dako | P026002-2 |
| Rat monoclonal anti-HA-Peroxidase | Roche | 12013819001 |
| Mouse monoclonal anti-CAp24 HIV-1 | NIBSC | EVA366 |
| Rabbit polyclonal anti-MAp17 HIV-1 | NIH AIDS Research and Reference Reagent Program | ARP-4811 |
| Rabbit polyclonal anti-Vpu HIV-1 | NIH AIDS Research and Reference Reagent Program | ARP-969 |
| Rabbit polyclonal anti-Vpr HIV-1 | NIH AIDS Research and Reference Reagent Program | ARP-3951 |
| Human monoclonal anti-SUgp120 HIV-1 | NIH AIDS Research and Reference Reagent Program | ARP-1476 |
| Mouse monoclonal anti-GAPDH | Santa Cruz Biotechnology | sc-47724 |
| Mouse monoclonal anti-tubulin | Sigma-Aldrich | T9026 |
| Mouse monoclonal anti-CD71 (TF-R) | Santa Cruz Biotechnology | sc-65882 |
| Rabbit polyclonal anti-ATG5 | Cell Signaling Technology | 12994S |
| Rabbit polyclonal anti-Beclin1 | Santa Cruz Biotechnology | sc-11427 |
| rabbit polyclonal anti-p62 | Life Technologies | PA5-27247 |
| Rabbit monoclonal anti-LC3A | Cell Signaling Technology | 4599T |
| Rabbit polyclonal LC3B | Novus Biologicals | NB600-1384 |
| Rabbit monoclonal anti-GABARAP | Cell Signaling Technology | 13733 T |
| Rabbit monoclonal anti-GABARAPL1 | Cell Signaling Technology | 26632 T |
| Rabbit monoclonal anti-GABARAPL2 | Cell Signaling Technology | 14256 T |
| Mouse monoclonal anti-TSG101 | BD Biosciences | 612697 |
| Human IgG1κ control | Sigma-Aldrich | I5154 |
| Human polyclonal anti-HIV-1 SF2 p24 | NIH AIDS Research and Reference Reagent Program | ARP-4250 |
| Rabbit serum | Sigma-Aldrich | R9133 |
| Rabbit polyclonal anti-GABARAPL1 | Proteintech | 11010-1-AP |
| Rabbit IgG control | Cell Signaling Technology | 3900S |
| KC57-fluorescein isothiocyanate (FITC) | Beckman Coulter | 6604665 |

| Reagent/resource | Reference or source | Identifier or catalog number |
|---|---|---|
| **Oligonucleotides and other sequence-based reagents** | | |
| Primer name | Sequence 5'-3' | |
| Viral gRNA (Jablonski et al J Virol 2009 – PMID : 19004959) | US Fw: 5'-TTCTTCAGAGCAGACCAGAGC-3' | |
| | US Rev: 5'-GCTGCCAAAGAGTGATCTGA-3' | |
| LC3A | LC3A Fw: 5'-CAGCACCCCAGCAAAATC-3' | |
| | LC3A Rev: 5'-TCTTTCTCCTGCTCGTAGA-3' | |
| LC3B | LC3B Fw: 5'-AGCAGCATCCAACCAAAATC-3' | |
| | LC3B Rev: 5'-TGTGTCCGTTCACCAACAG-3' | |
| LC3C | LC3C Fw: 5-'CCCAAGCGTCAGACCCTTC-3' | |
| | LC3C Rev: 5'-GCTCCGGATGATGCTGAG-3' | |
| GABARAP | GB Fw: 5'-TGCCGGTGATAGTAGAAA-3' | |
| | GB Rev: 5'-GGTGTTCCTGGTACAGCT-3' | |
| GABARAPL1 | L1 Fw: 5'-TGCCCTCTGACCTTACTG-3' | |
| | L1 Rev: 5'-AGTCTTCCTCATGATTGTC-3' | |
| GABARAPL2 | L2 Fw: 5'-TCGCTGGAACACAGATG-3' | |
| | L2 Rev: 5'-TGTCCCATAGTTAGGCTG-3' | |
| 7SL | 7SL Fw: 5'-GCCTGTAGTCCCAGCTACTC-3' | |
| | 7SL Rev: 5'-CCGAACTTAGTGCGGACACC-3' | |
| 7SK | 7SK Fw: 5'-CCCTGCTAGAACCTCCAAAC-3' | |
| | 7SK Rev: 5'-AAGAAAGGCAGACTGCCAC-3' | |
| Human actin | Act hm Fw: 5'-CCCTGGACTTCGAGCAAGAG-3' | |
| | Act hm Rev: 5'-ACTCCATGCCCAGGAAGGAA-3' | |
| Mouse actin | Act ms Fw: 5'-CCAGGTCATCACTATTGGCAAC-3' | |
| | Act ms Rev: 5'-TTGGCATAGAGGTCTTTACGGAT-3' | |
| FKBP4 | FKBP4 Fw: 5'-AAGGCGTGCTGAAGGTCAT-3' | |
| | FKBP4 Rev: 5'-CCAGCCAGTGTAGTGGACAA-3' | |
| Others | | |
| RNAscope-probe HS-HIV | Bio-Techne | 311921-C1 |
| **Chemicals, enzymes and other reagents** | | |
| DMEM, high glucose, GlutaMAX™ Supplement | Gibco, Life Technologies | 61965059 |
| RPMI 1640 medium, GlutaMAX™ Supplement | Gibco, Life Technologies | 61870044 |
| Fetal calf serum (FCS) qualified | Gibco, Life Technologies | 10270106 |
| Dulbecco's phosphate-buffered saline (PBS) | Gibco, Life Technologies | 14190-169 |
| Trypsin | Promega | V511A |
| Sodium dodecyl sulfate (SDS) | Sigma-Aldrich | L4509 |
| Tris | GE Healthcare-Cytiva | 17-1321-01 |
| Tris(2-carboxyethyl) phosphine hydrochloride (TCEP) | Sigma-Aldrich | C4706 |

| Reagent/resource | Reference or source | Identifier or catalog number |
|---|---|---|
| 2-Chloroacetamide (CAA) | Sigma-Aldrich | 22788 |
| Trifluoroacetic acid (TFA) | Sigma-Aldrich | 302031 |
| Acetonitrile (ACN) | Sigma-Aldrich | 1000292500 |
| Formic acid | Sigma-Aldrich | F0507 |
| Antibiotic-antimycotic cocktail | Gibco, Life Technologies | 15240062 |
| Human IL-2 | Miltenyi Biotec | 130-097-748 |
| Bafilomycin A1 | Sigma-Aldrich | B1793 |
| AMD3100 | Sigma-Aldrich | A5602 |
| Lymphocytes separation medium | Eurobio scientific | CMSMSL01-01 |
| CD4 + T Cell Isolation Kit, human | Miltenyi Biotec | 130-096-533 |
| T Cell TransAct™, human | Miltenyi Biotec | 130-111-160 |
| Lipofectamine RNAiMAX | Life Technologies | 13778075 |
| Lipofectamine LTX with PLUS Reagent | Life Technologies | 15338100 |
| Polyethylenimine (PEI) | Polysciences | 23966 |
| S-Trap Micro Spin Column | Protifi, Farmingdale, NY, USA | C02-micro-80 |
| HIV-1 Cap24 ELISA kit | Revvity | NEK050B001KT |
| Galacto-Star™ β-Galactosidase Reporter Gene Assay System for Mammalian Cells | Applied Biosystems | T1014 |
| Tris hydrochloride solution pH = 8 | Sigma-Aldrich | T3038 |
| Sodium chloride solution | Sigma-Aldrich | S5150 |
| Triton X-100 | Sigma-Aldrich | T8787 |
| Sodium deoxycholate (DOC) | Sigma-Aldrich | D6750 |
| EDTA -free complete protease Inhibitors | Roche | 05056489001 |
| Protein Assay Dye Reagent | Bio-Rad | 5000006 |
| Laemmli 2X concentrate | Sigma-Aldrich | S3401 |
| Hydrophobic polyvinylidene difluoride membranes (PVDF, 0.45 μm) | Millipore | IPVH00010 |
| Amersham ECL Select Western blotting detection reagent | GE Healthcare | RPN2235 |
| Tris base | Euromedex | 200923-A |
| Sodium chloride | Euromedex | 1112-A |
| Tween 20 | Sigma-Aldrich | P1379 |
| BSA | Euromedex | 04-100-812-D |
| Tris hydrochloride solution pH = 7.4 | Sigma-Aldrich | T2194 |
| Magnesium chloride solution ($MgCl_2$) | Sigma-Aldrich | 63069 |
| Turbo DNase I | Ambion | AM2239 |
| RNase | Sigma-Aldrich | R6513 |

| Reagent/resource | Reference or source | Identifier or catalog number |
| --- | --- | --- |
| HA-trap magnetic agarose beads | ChromoTek | atma-20 |
| Amicon units | Sigma-Aldrich | UFC910024 |
| Dynabeads protein G | Life Technologies | 10004D |
| Hepes | Gibco, Life Technologies | 11560496 |
| DL-Dithiothreitol (DTT) | Sigma-Aldrich | D9779 |
| Dynabeads protein A | Life Technologies | 10002D |
| TRIzol LS reagent | Life Technologies | 10296028 |
| GlycoBlue | Life Technologies | AM9516 |
| Subtilisin | Sigma-Aldrich | 85968 |
| PMSF | Sigma-Aldrich | P7626 |
| Reliaprep RNA Cell Miniprep System kit | Promega | Z6012 |
| UltraPure Distilled water DNase/RNase free | Life Technologies | 10977049 |
| DNase I | Life Technologies | AM2239 |
| High-capacity cDNA Reverse Transcription Kit | Applied Biosystems | 4368814 |
| SYBR Green Supermix | Bio-Rad | 1725275 |
| IGEPAL CA-630 (NP40) | Sigma-Aldrich | I3021 |
| Urea | Sigma-Aldrich | U5378 |
| Sodium dodecyl sulfate solution | Sigma-Aldrich | 15553-035 |
| Proteinase K | Roche | P2308 |
| Puromycine dichlorhydrate | Life Technologies | A1113803 |
| Digitonin | Calbiochem | 300410 |
| RNAscope™ Multiplex Fluorescent Reagent Kit v2 | Bio-Techne | 323100 |
| RNAscope™ Probe Diluent - RNAscope™ probe diluent | Bio-Techne | 300041 |
| Opal 570 Reagent Pack | Akoya Biosciences, Inc | FP1488001KT |
| Prolong Diamond Antifade Mountant | Life Technologies | P36970 |
| **Software** | | |
| GraphPad Prism 6 | https://www.graphpad.com/ | |
| IMARIS | https://imaris.oxinst.com/ | |
| Fiji | https://fiji.sc/ | |
| DIA-NN version 1.8.1 | Demichev et al, (2020) | |
| R | https://www.r-project.org/ | |
| Photoshop CS5 | Adobe | |
| **Other** | | |
| μ-Slide VI 0.4 | IBIDI | 80606 |
| ACD HybEZ™ II Hybridization System | Bio-Techne | 321720-R |
| Real-time PCR detection system | Roche | LightCycler® 480 |
| Stratalinker 254-nm UV crosslinker 2400 | Stratagene | SS-UV2400 |

| Reagent/resource | Reference or source | Identifier or catalog number |
| --- | --- | --- |
| Dionex U3000 RSLC nano-LC- system | Thermo Scientific | N/A |
| TIMS-TOF Pro mass spectrometer | Bruker Daltonik GmbH | N/A |
| Microscope IXplore spinning disk | Olympus | N/A |

## Cell lines

HeLa, HEK293T, and HeLa P4R5 cell were grown in DMEM (Dulbecco's modified Eagle's medium) with GlutaMAX supplemented with 10% decomplemented FCS (fetal calf serum) (Gibco, Life Technologies), and were maintained at 37 °C in a 5% $CO_2$ incubator. Parental, CRISPR-Cas9 knockout cells for LC3A, LC3B, and LC3C (LC3 TKO), and for GABARAP, GABARAPL1, and GABARAPL2 (GABARAP TKO) were a gift from Lazarou's lab (Nguyen et al, 2021). Parental and CRISPR-Cas9 knockout cells for ATG5 (ATG5−/− 92#3) were previously described (Judith et al, 2023). Cells were tested for mycoplasma contamination.

## Purification, culture, and characterization of CD4 + T cells

CD4 + T cells were obtained from HIV-seronegative human buffy coats. Peripheral blood mononuclear cells (PBMCs) were purified by density centrifugation using lymphocytes separation medium (density, 1.077 g/mL; Eurobio Scientific) at 400×*g* for 30 min at 20 °C. CD4 + T cells were negatively selected from PBMCs using magnetic-activated cell sorting (MACS) CD4 + T Cell Isolation Kit (Miltenyi Biotec) according to the manufacturer's recommendations. CD4 + T cells were then activated and expanded using T Cell TransAct reagent (Miltenyi Biotec) according to the manufacturer's recommendations. After 3 days of stimulation, T Cell TransAct reagent was removed and CD4 + T cells were cultured in RPMI 1640 medium (Gibco, Life Technologies) supplemented with 10% decomplemented FCS, 1% antibiotic-antimycotic cocktail and 30 IU/mL human IL-2 (Miltenyi Biotec). Three days after initial stimulation, the purity and the activation of CD4 + T cells were assessed by flow cytometry analysis of the following cell surface markers: CD3, CD4, CD8, CD25, and CD69.

## Small interfering RNA and transfection

Cells were transfected with relevant small interfering RNA (siRNA) oligonucleotide using Lipofectamine RNAiMAX (Life Technologies), according to the reverse transfection procedure described in the manufacturer's recommendations. The final concentration of siRNA oligonucleotides was 10 to 20 nM. SiRNA sequence targeting LC3A (5'-GGCTTCCTCTATATGGTCTACGCCT-3', positions 491–515 for variant 1 and 437–461 for variant 2), LC3C (5'-GCTTGGCAATCAGACAAGAGGAAGT-3', positions 128-152), GABARAP (5'-GAGGGCGAGAAGATCCGAAAGAAAT-3', positions 153-177), GABARAPL1 (5'-GAGGACAATCATGAG-GAAGACTATT-3', positions 526-550), GABARAPL2 (5'-GATCTTCCTGTTTGTGGATAAGACA-3', positions 361-385).

The siRNAs targeting LC3B (L-012846-00) were purchased from Dharmacon. The siRNA 5'-TGGTTTTACATGTCGACTAA-3' (D-001810-01 from Dharmacon) was used as a negative control (referred to as siRNA Control).

## Antibodies

The following antibodies were used for immunoblotting: mouse monoclonal anti-CAp24 HIV-1 (NIBSC; EVA366), rabbit polyclonal anti-MAp17 HIV-1 (NIH; ARP-4811), rabbit polyclonal anti-Vpu HIV-1 (NIH; ARP-969), rabbit polyclonal anti-Vpr HIV-1 (NIH; ARP-3951), human monoclonal anti-SUgp120 HIV-1 (NIH; ARP-1476), rat monoclonal anti-HA-Peroxidase (Roche; 12013819001), mouse monoclonal anti-GAPDH (Santa Cruz; sc-47724), mouse monoclonal anti-tubulin (Sigma-Aldrich; T9026) mouse monoclonal anti-CD71 (TF-R) (Santa Cruz; sc-65882), rabbit polyclonal anti-ATG5 (Cell Signaling Technology; 12994S), rabbit polyclonal anti-Beclin1 (Santa Cruz; sc-11427), rabbit polyclonal anti-p62 (Life Technologies; PA5-27247), rabbit monoclonal anti-LC3A (Cell Signaling Technology; 4599 T; This antibody may also react with LC3B), rabbit polyclonal LC3B (Novus Biologicals; NB600-1384), rabbit monoclonal anti-GABARAP (Cell Signaling Technology; 13733 T), rabbit monoclonal anti-GABARAPL1 (Cell Signaling Technology; 26632 T), rabbit monoclonal anti-GABARAPL2 (Cell Signaling Technology; 14256 T), and mouse monoclonal anti-TSG101 (BD Biosciences; 612697). Secondary antibodies against the mouse and the rabbit immunoglobulin G's coupled to HRP (Dako) were used for immunoblotting experiments.

The following antibodies were used for IP: human monoclonal anti-SUgp120 HIV-1 (NIH; ARP-1476), human IgG1κ control (Sigma-Aldrich; I5154), human polyclonal anti-HIV-1 SF2 p24 (NIH; ARP-4250), rabbit serum (Sigma-Aldrich; R9133), rabbit polyclonal anti-GABARAPL1 (Proteintech; 11010-1-AP), and rabbit IgG control (Cell Signaling; 3900S).

## Mammalian expression vectors and transfection

The ORF of human GABARAP, GABARAPL1, and GABARAPL2 were cloned into the pCMV-HA vectors. Transfections of HeLa cells with expression vectors were performed using Lipofectamine LTX with PLUS Reagent (Life Technologies), following the manufacturer's instructions.

## Indirect measurement of autophagy activity

Parental, LC3 TKO and GABARAP TKO HeLa cells were incubated in full medium or EBSS (Earle's Balanced Salt Solution; 5.56 mM D-glucose, 23.08 mM NaCl, 5.37 mM KCl, 1.82 mM CaCl$_2$, 0.81 mM MgSO$_4$, 0.99 mM Na$_2$HPO$_4$, 13.10 mM NaHCO$_3$) for amino acid depletion without or with Bafilomycin A1 (Sigma-Aldrich) for 2 h. Cell lysates were analyzed by western blot for the detection of LC3A/B, LC3B, GABARAP, GABARAPL1, GABARAPL2, and GAPDH proteins.

## Viral stocks production

Stocks of VSVg pseudotyped wild-type (WT) or NL4-3 (MA/YFP) HIV-1, a gift from Dr P. Bienasz (Jouvenet et al, 2006), were obtained by transfection of HEK293T cells with HIV-1 proviral DNA along with a VSVg expression vector (pMD.G) and polyethylenimine (PEI) (Polysciences). Twenty-four hours after transfection, cells media were removed, and cells were cultured for an additional 24 h in fresh media. Supernatants were then collected and 0.45-μm-filtered. Viral titers were determined by infection of HeLa cells with serial dilutions of the viral stocks for 24 h, followed by flow cytometry analysis of CAp24 antigen expression on fixed and permeabilized cells labeled with KC57-fluorescein isothiocyanate (FITC) (Beckman Coulter).

## Retroviral vectors production

The ORF of human GABARAPL1 and its non-lipidated mutated variants (GABARAPL1 G116A) were cloned into the pOZ-FH-N retroviral vectors (Sobhian et al, 2010). Stocks of each retroviral vectors were obtained by transfection of HEK293T cells with corresponding pOZ-FH-N DNA along with a VSV-G expression vector (pMD.G), pCMV-Gag-Pol vector and polyethylenimine (PEI) (Polysciences). Twenty-four hours after transfection, cells media were removed, and the cells were cultured for an additional 24 h in fresh media. Supernatants containing retroviral vectors were then collected and 0.45-μm-filtered.

## Sample preparation for proteomic analyses

Cell pellets and viral particles were solubilized in lysis buffer (2% SDS, 100 mM Tris-HCl, pH 8.5) and boiled for 5 min at 95 °C. Samples were also reduced and alkylated (10 mM TCEP, 50 mM CAA). Thirty μg of each cell extracts and the whole samples of viral particles were digested using trypsin (Promega), and the S-Trap Micro Spin Column was used according to the manufacturer's protocol (Protifi, Farmingdale, NY, USA). Peptides were then speed-vacuum dried.

## Liquid chromatography-coupled mass spectrometry analysis (nLC-MS/MS)

nLC-MS/MS analyses were performed on a Dionex U3000 RSLC nano-LC system coupled to a TIMS-TOF Pro mass spectrometer (Bruker Daltonik GmbH, Bremen, Germany). After drying, cell peptides were solubilized in 30 μl of 0.1% TFA containing 10% acetonitrile (ACN) and viral particles in 10 μl of the same solution. Two μl and 1 μl were respectively loaded from cells and viral particles samples, concentrated and washed for 3 min on a C18 reverse phase column (5-μm particle size, 100 Å pore size, 300-μm inner diameter, 0.5 cm length, from Thermo Fisher Scientific). Peptides were separated on an Aurora C18 reverse phase resin (1.6-μm particle size, 100 Å pore size, 75-μm inner diameter, 25 cm length mounted to the Captive nanoSpray Ionization module, from IonOpticks, Middle Camberwell Australia) with a respectively 4 and 1 h run time for cells and viral particles samples with a gradient ranging from 98% of solvent A containing 0.1% formic acid in milliQ-grade H$_2$O to 35% of solvent B containing 80% acetonitrile, 0.085% formic acid in mQH$_2$O.

The mass spectrometer acquired data throughout the elution process and operated in DIA PASEF mode with a 1.38 s/cycle, with timed ion mobility spectrometry (TIMS) enabled and a data-independent scheme with full MS scans in parallel accumulation

and serial fragmentation (PASEF). Ion accumulation and ramp time in the dual TIMS analyzer were set to 100 ms each, and the ion mobility range was set from $1/K0 = 0.632$ to $1.43\,Vs\,cm^{-2}$. Precursor ions for MS/MS analysis were isolated in positive polarity with PASEF in the 400–1.200 m/z range by synchronizing quadrupole switching events with the precursor elution profile from the TIMS device.

## Identification and quantification of proteins detected by nLC-MS/MS

The mass spectrometry data were analyzed using DIA-NN version 1.8.1 (Demichev et al, 2020). The database used for in silico generation of the spectral library was a concatenation of Human and HIV-1 NL4-3 sequences from the SwissProt (release 2022-05) and NCBI databases, respectively, and a list of contaminant sequences. Oxidation of methionines was set as a variable modification, carbamidomethylation of cysteines was set as a permanent modification, and one trypsin miscleavage was allowed. The precursor false discovery rate (FDR) was kept below 1%. The "match between runs" (MBR) option was not allowed. The mass spectrometry proteomics data have been deposited to the ProteomeXchange Consortium via the Pride partner repository with the dataset identifiers PXD057162 for cell lysates and PXD057129 for viral particles.

Quantification analysis was done using a home R script. Log2 of proteins intensities were calculated and then a one-tailed paired Student $t$-test was done on proteins showing 70% of valid value in at least one condition.

A total of 2919 human proteins are identified in HIV-1 vs mock preparations. These proteins are present in four viral preparations. Among these 2919, 1838 proteins are enriched with a fold change equal or superior to 1.3 (in four viral preparations and in at least three mock preparations), and 46 proteins are exclusively detected in four HIV-1 preparations, compared to mock preparations. Among the 2919 proteins, the ones identified in four viral preparations and in 1 or 2 mock preparations (4 vs 1, and 4 vs 2) are considered as non-statistically evaluable.

## HIV-1 production assay

CD4 + T cells were infected by spinoculation with VSVg pseudo-typed NL4-3 HIV-1 for 2 h at an MOI of 0.5. Forty-eight hours after infection, supernatants were collected, filtered and ultracentrifuged on a 20% sucrose cushion for 1 h 30 at 150,000×g. Cell lysates and supernatants were analyzed by western blotting.

HeLa were transfected with WT NL4-3 HIV-1, Udel NL4-3 HIV-1, ΔVif NL4-3 HIV-1, ΔVpr NL4-3 HIV-1, and ΔNef NL4-3 HIV-1 proviral DNA. Twenty-four hours after transfection, the media was removed and replaced with fresh media for an additional 24 h. Supernatants were then collected, 0.45 μm-filtered and ultracentrifuged on a 20% sucrose cushion for 1 h at 150,000×g. Cell lysates and supernatants were analyzed by western blotting.

In a single round of infection, HeLa cells were transfected with siRNA (10-20 nM) as described above. Forty-eight hours after, siRNA-treated HeLa cells were infected with VSVg-pseudotyped NL4-3 HIV-1 for 2 h at a multiplicity of infection (M.O.I.) of 0.5. Thirty-two hours after infection, the media was removed and replaced with fresh media for an additional 16 h. Supernatants were then collected, 0.45 μm-filtered, ultracentrifuged on a 20% sucrose cushion for 1 h at 150,000×g, and used for HIV-1 CAp24 quantification by ELISA (PerkinElmer). HIV-1 release index corresponds to the ratio of released CAp24 to cell-associated CAp24. The infectivity of the released virus was determined as described below. Cell lysates and supernatants were analyzed by western blotting.

HeLa, HeLa ATG5−/− 92#3, HeLa LC3 TKO and HeLa GABARAP TKO cells were transfected with WT NL4-3 HIV-1 proviral DNA. Twenty-four hours after transfection, the media was removed and replaced with fresh media for an additional 24 h. Supernatants were then collected, 0.45 μm-filtered, ultracentrifuged on a 20% sucrose cushion for 1 h at 150,000×g, and used for HIV-1 CAp24 quantification by ELISA (PerkinElmer). HIV-1 release index corresponds to the ratio of released CAp24 to cell-associated CAp24. Cell lysates and supernatants were analyzed by western blotting. Total cellular and supernatant RNA were extracted to be analyzed by RT-qPCR as described below. The infectivity of the released virus was determined by infection of the indicator cells HeLa P4R5 in a standardized 96-well titration assay by lumino-metric analysis of β-galactosidase activity (Galacto-Star System, Applied Biosystems) following the manufacturer's instructions, then normalized by the quantity of released CAp24. To analyze the viral production from a single round of infection, the medium was changed 24 h after infection with fresh medium supplemented with 10 μM AMD3100 (Sigma-Aldrich; A5602).

HeLa GABARAP triple knockout (TKO) cells were transduced with pOZ empty vector (pOZ), and pOZ retroviral vectors encoding either HA-tagged wild-type (pOZ-HA-L1 WT) or non-lipidated mutant form (pOZ-HA-L1 G116A) of GABARAPL1 proteins. Twenty-four hours post-transduction, the medium was replaced with fresh medium containing 1 μg/mL puromycin for selection. After 7 days of selection, puromycin-containing medium was removed, and the resulting stable cell lines, expressing wild-type GABARAPL1 protein or its corresponding C-terminal glycine-to-alanine mutant, were validated by western blotting. Parental, GABARAP TKO, HA-GABARAPL1 WT, or G116A trans-complemented TKO cell lines were then infected with VSV-G pseudotyped NL4-3 HIV-1 for 2 h at a multiplicity of infection (MOI) of 0.3. Forty-eight hours post-infection, supernatants were collected, filtered through 0.45 μm filters, and subjected to ultracentrifugation through a 20% sucrose cushion at 150,000×g for 1 h. Viral supernatants were used for HIV-1 capsid p24 quantification by ELISA (PerkinElmer). The RNA release index was calculated as the ratio of HIV-1 gRNA (normalized to mouse actin mRNA) present in supernatant samples to HIV-1 gRNA (normalized to actin mRNA) present in cellular samples. RNA packaging was calculated as the ratio of HIV-1 gRNA (normalized to mouse actin mRNA) in supernatant samples to the amount of released CAp24, as quantified by ELISA.

## Western blotting

Cell lysates were lysed in ice-cold DOC buffer (10 mM Tris, pH 8, 150 mM NaCl, 1 mM EDTA, 1% Triton X-100, and 0.1% DOC 10%) containing complete protease inhibitor cocktail (Roche). Lysates were cleared by centrifugation for 25 min at 18,000×g. The protein concentrations were determined using a Bradford protein assay (Bio-Rad), and equal amounts of protein for each sample were used for

the following steps. Cell lysates and supernatants were subjected to SDS-PAGE gels. Laemmli 2X concentrate (Sigma-Aldrich) has been used as a sample buffer for reducing and loading protein samples in SDS-PAGE. Proteins were then transferred onto hydrophobic polyvinylidene difluoride membranes (PVDF, 0.45 μm, Millipore), followed by blocking in milk buffer (Tris-buffered saline [TBS] [0.5 M Tris pH 8.4, 9% {wt/vol} NaCl], 5% [wt/vol] nonfat dry milk, 0.05% [vol/vol] Tween 20) for 1 h at room temperature. Membranes were incubated overnight at 4 °C with the appropriate primary antibodies in milk buffer or BSA buffer (Tris-buffered saline [TBS] [0.5 M Tris pH 8.4, 9% {wt/vol} NaCl], 3% [wt/vol] BSA, 0.05% [vol/vol] Tween 20). Blots were washed with TBS containing 0.05% (vol/vol) Tween 20 and incubated with appropriate HRP-conjugated secondary antibodies in milk or BSA buffer for 1 h at RT. After washing, protein bands were detected by using Amersham ECL Select Western blotting detection reagent (GE Healthcare).

## Co-immunoprecipitation (co-IP)

Parental HeLa cells were co-transfected with WT NL4-3 HIV-1 proviral DNA, and with pCMV-HA-LC3B, -GABARAP, -GABARAPL1, -GABARAPL2, or pCMV-HA-mock as a control. Twenty-four hours after transfection, cells were washed with 1X PBS and incubated with the Lysis buffer (20 mM Tris-HCl pH 7.4, 150 mM NaCl, 5 mM EDTA, 0.4% Triton, and 6 mM MgCl$_2$) with complete protease inhibitor cocktail (Roche), for 30 min on ice. Samples are digested with 60 U of Turbo DNase I (Ambion) during 10 min at 37 °C and centrifuged 5 min at 100×$g$ at 4 °C and the supernatants are recovered. Each sample (3 μg) was treated, or not, with 15 U of RNase (Sigma-Aldrich; R6513) for 10 min at 30 °C (30 μl RNase/DNase-free H$_2$O is added for mock-treated). After 5 min on ice, supernatants are recovered. Aliquots (input) were kept for subsequent protein analysis before IP. IPs were performed on equal amounts of proteins for each sample by incubating whole cell extracts for 1 h 30 at 4 °C with HA-trap magnetic agarose beads (ChromoTek). The beads were washed five times with lysis buffer, and proteins were eluted in 2X Laemmli (Sigma-Aldrich). Relative amounts of Pr55Gag and HA-GABARAPL1 were quantified using Fiji software.

## HIV-1 particle immunoprecipitation (IP) assay

Parental HeLa or GABARAP TKO cells were infected with VSVg pseudotyped NL4-3 HIV-1 for 2 h at an MOI of 1. Twenty-four hours after infection, the media was removed and replaced with DMEM without FCS for an additional 24 h. Supernatants were then collected, 0.45 μm-filtered and the viruses were concentrated using Amicon units (Sigma-Aldrich; UFC910024) with 20 min of centrifugation at 3000× $g$. The viruses were precleared with Dynabeads protein G (Life Technologies) for 1 h at 4 °C. Aliquots of supernatant were kept for subsequent protein and RNA analysis before IP. IPs were performed on equal amounts of viruses for each sample by incubating purified supernatants overnight at 4 °C with human monoclonal anti-SUgp120 HIV-1 or human IgG1κ control. The virus-antibody complexes were coupled to Dynabeads protein G for 1 h at room temperature. The beads were then washed three times with the washing buffer (20 mM HEPES, pH 7.4, 150 mM NaCl, and 1 mM EDTA) with a complete protease inhibitor cocktail, and proteins were eluted in 2X Laemmli (Sigma-Aldrich).

For RNA analysis, beads were washed three times with the washing buffer (20 mM HEPES, pH 7.4, 150 mM NaCl, and 1 mM EDTA) and three times with the washing buffer without EDTA. Beads were then incubated with lysis buffer (1X PBS, 0.5% NP40, 6 mM MgCl$_2$, and 2 mM dithiothreitol) containing complete protease inhibitor cocktail (Roche) for 20 min on ice. Supernatants were recovered and RNAs from IPs and precleared viruses were extracted using TRIzol LS reagent (Life Technologies) and precipitated with ethanol:isopropanol (1:1 ratio) and GlycoBlue (15 μg/μl, Life Technologies). RNAs were quantified by RT-qPCR and normalized to precleared virus fractions and control immunoprecipitations.

## Subtilisin treatment of HIV-1 particles

HeLa cells were infected with VSVg pseudotyped NL4-3 HIV-1 for 2 h at an MOI of 1. Twenty-four hours after infection, the media were removed and replaced with fresh media for an additional 24 h. Cell lysates were then recovered and supernatants were collected, 0.45 μm-filtered, ultracentrifuged on a 20% sucrose cushion for 1 h 30 at 150,000×$g$ and used for HIV-1 CAp24 quantification by ELISA (PerkinElmer). Control and treated supernatants were performed on equal amounts of HIV-1 CAp24 protein. Supernatants were then treated with 1 mg/mL subtilisin (Sigma-Aldrich; 85968) for 5 min at room temperature. Reaction was stopped with 5 mM PMSF (Sigma-Aldrich; P7626). Samples were lysed with 5X Laemmli and analyzed by western blotting.

## RNA extraction and quantitative RT-PCR

Total cellular RNA was extracted using the Reliaprep RNA Cell Miniprep System kit (Promega; Z6012) following the manufacturer's instructions. Total supernatant RNA was mixed with TRIzol LS reagent (Life Technologies), 5 ng of spike-in (mouse actin RNA) were added, and RNA precipitate was purified following the manufacturer's instructions. About 15 μg of GlycoBlue (15 μg/μl, Life Technologies) were used to visualize the RNA pellet. Purified RNA was resuspended in 20 μl of UltraPure Distilled water DNase/RNase free (Life Technologies). For each cellular sample, 500 ng to 2 μg of total RNA were subjected to DNase I treatment (TURBO DNase; Life Technologies; cat AM2239), and 100 to 150 ng of total RNA for supernatant samples. cDNA synthesis was performed using the High-capacity cDNA Reverse Transcription Kit (Applied Biosystems; 4368814). The different RNA levels were assayed using SYBR Green Supermix (Bio-Rad; 1725275) in a real-time PCR detection system (LightCycler® 480). The PCR conditions and cycles were as follows: an initial DNA denaturation at 95 °C for 5 min, followed by 45 cycles of amplification (denaturation: 95 °C for 10 s, annealing: 63 °C for 10 s, and extension: 72 °C for 10 s), followed by a melting-curve analysis cycle. Each point was performed in technical duplicate. The relative abundance of different HIV-1 RNA species, using specific primers (Reagents and tools table), were calculated by the comparative ΔΔCt method, normalizing to the housekeeping product actin mRNA or 7SL mRNA for cellular samples of HeLa cells, FKBP4 mRNA in LT CD4+ cells and to mouse actin mRNA for supernatant samples. The RNA release index was calculated as the ratio of HIV-1 gRNA (normalized to mouse actin mRNA) present in supernatant samples to HIV-1 gRNA (normalized to actin mRNA) present in cellular

samples. RNA packaging was calculated as the ratio of HIV-1 gRNA (normalized to mouse actin mRNA) in supernatant samples to the amount of released CAp24, as quantified by ELISA.

## RNA immunoprecipitation (RIP) assay

Parental HeLa and GBRP TKO cells were infected with VSVg pseudotyped HIV-1 viruses at an MOI of 1. Forty-eight hours later, infected cells were rinsed, covered by ice-cold 1X phosphate-buffered saline (PBS) and irradiated at 400 and 200 mJ/cm$^2$ subsequently using a Stratalinker 254-nm UV crosslinker 2400. Cells were scraped and lysed 20 min on ice in Lysis buffer (1X PBS, 0.5% NP40, 6 mM MgCl$_2$, and 2 mM dithiothreitol) containing complete protease inhibitor cocktail (Roche), then digested with 60 U of Turbo DNase I (Ambion) during 10 min at 37 °C and centrifuged 5 min at 1000 rpm at 4 °C. Aliquot of supernatant was kept for subsequent protein and RNA analysis before IP (input). The rest was incubated overnight with Protein A Dynabeads (Life Technologies) previously incubated with human polyclonal anti-HIV-1 SF2 p24 (NIH; ARP-4250), rabbit serum (Sigma-Aldrich; R9133), rabbit polyclonal anti-GABARAPL1 (Proteintech; 11010-1-AP) and rabbit IgG control (Cell Signaling; 3900S). Dynabeads were washed four times with Wash buffer (5X PBS, 1% NP40, 1% Sodium deoxycholate, 6 mM MgCl$_2$, 2 mM dithiothreitol, 0.1% SDS, and 4 M urea) containing complete protease inhibitor cocktail (Roche), then three times with lysis buffer. Beads were split into two for western blot analysis and RT-qPCR analysis. For western blotting, proteins were eluted in 2X Laemmli (Sigma-Aldrich). For RT-qPCR analysis, beads were subjected to proteinase K digestion in PK buffer (100 mM Tris-HCl, pH 7.4, 50 mM NaCl, and 10 mM EDTA) containing 400 µg of proteinase K (Roche). The input fraction was digested similarly for RNA extraction. RNAs from RIP and inputs were extracted using TRIzol LS reagent (Life Technologies), and precipitated with ethanol:isopropanol (1:1 ratio) and GlycoBlue (15 µg/µL, Life Technologies). RNAs were quantified by RT-qPCR and normalized to input and control immunoprecipitations.

## Subcellular fractionation

Parental HeLa and GABARAP TKO cells were infected with VSVg pseudotyped HIV-1 viruses at an MOI of 1 for 48 h. On the day of fractionation, cells were washed once with ice-cold PBS and then scraped. Cell suspensions were pelleted by centrifugation at 500×$g$ for 5 min at 4 °C. The supernatant was discarded, and the cell pellet was resuspended in 2 ml of fractionation buffer (20 mM Tris-HCl, pH 7.5, 300 mM NaCl, 5 µg/ml digitonin, and 1X complete protease inhibitor). Cell suspension was left for 20 min on ice with occasional gentle mixing, and then cells were homogenized by passing through a Dounce homogenizer. Nuclear pellets were pelleted by centrifugation at 2000×$g$ for 10 min. A sample of the supernatant was set aside as the input, and the remainder was taken for an ultracentrifuge spin at 100,000×$g$ for 1 h at 4 °C. The resultant supernatant produces the cytosol fraction, and the membrane fraction is contained in the pellet fraction. Pellet fractions were washed in fractionation buffer without digitonin and were taken for a second spin at 100,000×$g$ for 1 h at 4 °C. For western blotting and RT-qPCR analysis, equal volumes of each fraction were used. RNAs were extracted using TRIzol LS reagent (Life Technologies), and precipitated with ethanol:isopropanol (1:1 ratio) and GlycoBlue (15 µg/µL, Life Technologies). RNAs were quantified by RT-qPCR and normalized to 7SL mRNA.

## In situ HIV-1 gRNA detection

HIV-1 gRNA in cells was probed using RNAscope reagents (Bio-Techne) using the manufacturer's protocol. Briefly, cells were seeded onto µ-Slide VI 0.4 (IBIDI) and infected for 24 h with VSV-G-pseudotyped HIV-1 WT (MA-YFP) at a MOI of 0.5. Following fixation, cells were dehydrated by removal of PBS and sequential replacement with 50, 70% then 100% ethanol, incubating the samples for 5 min at room temperature in each solution. Ethanol solutions were prepared as volume-to-volume ratios in ultrapure RNase-DNAse-free water (Lifetechnology). The 100% ethanol was replaced with fresh 100% ethanol and incubated at room temperature for a final 10 min. At this point IBIDI slide can be stored in 100% ethanol at −20 °C. To rehydrate cells, the sequence was reversed and the cells were incubated for 2 min at room temperature in each solution; the cells were not allowed to dry in air at any time during the process. Finally, 50% ethanol was replaced with PBS, and the cells were hydrated at room temperature for 10 min.

The manufacturer's protease Plus solution was diluted in PBS as appropriate prior to the experiment (1:15 dilution) and incubated in a humidified HybEZ oven (Bio-Techne) at 40 °C for 15 min. Protease solution was discarded and the slides were washed twice in PBS at room temperature for 2 min. Specific pre-designed RNAscope-probe HS-HIV (Bio-Techne, 311921-C1) that recognizes HIV-1 gRNA was diluted 1/16 in probe diluent (Bio-techne). Probes were allowed to hybridize with the samples in a humidified HybEZ oven at 40 °C for 2 h. The probes excess was then discarded and the slide washed twice using the manufacturer's wash buffer for 2 min. The probes were visualized by hybridizing with preamplifiers, amplifiers, and finally, a fluorescent label.

Pre-amplifier 1 (Amp 1) was hybridized to its cognate probe in a humidified HybEZ oven at 40 °C for 30 min. Samples were washed twice, then hybridized with amplifier 2 (Amp 2) in a humidified HybEZ oven at 40 °C for 30 min. After a further two washes, amplifier 3 (Amp 3) was hybridized in a humidified HybEZ oven at 40 °C for 15 min. Samples were washed twice, then HRP-C1 was hybridized in a humidified HybEZ oven at 40 °C for 15 min, then washed twice more. Fluorescent label OPAL 570 (Akoya) was diluted in TSA buffer (1/1000) and hybridized in a humidified HybEZ oven at 40 °C for 20 min, then washed twice.

The slide was incubated with the manufacturer's HRP blocker for 15 min in a humidified HybEZ oven at 40 °C. After a further two washes, the final step was to counter-stain nuclei with DAPI for 2 min at room temperature, wash twice with PBS and immediately mount the slides using Prolong Gold Antifade (Invitrogen).

## Imaging and imaging quantification

Cells were imaged using the Microscope IXplore spinning disk Olympus. We used the 60X plan-apochromat objective, with a numerical aperture of 1.42. In all experiments, images shown in individual panels were acquired using identical exposure times and scan settings and adjusted identically for brightness and contrast using Photoshop CS5 (Adobe).

Image stacks were acquired across the full cell volume with a Z-step size of 0.24 microns to enable three-dimensional (3D) analysis. The full 3D image volume was processed in IMARIS software. Individual infected cells were identified and segmented using the "Cells" module. Segmentation of nuclei, cytoplasm, and gRNA puncta enabled the extraction of individual measurements for each region. For each infected cell, the number of gRNA puncta was quantified in 3D and their localization (nuclear or cytoplasmic), was determined based on their spatial position relative to segmented compartments.

## Statistical analysis

No randomization or blinding was performed. Sample allocation to experimental groups was based on a predefined experimental layout and practical considerations. All experiments were conducted under the same standardized conditions to ensure reproducibility and reduce technical variability. All statistical analyses were performed using GraphPad Prism 6 software, as detailed in the figure legends. The type of test used (e.g., one-way ANOVA with Dunnett's or Sidak's multiple comparisons test, or unpaired Student's $t$-test) is specified for each figure. Data were presented as mean ± SEM. For each experiment, biological replicates ($n$) are clearly indicated in the figure legends. Normal distribution of the data were assumed based on the experimental design and previous experience with the system, but formal normality tests were not conducted. The variance was visually inspected and appeared similar between groups; where unequal variance or non-normal distribution was suspected, non-parametric alternatives or data normalization were considered. No data points were excluded unless clearly justified (e.g., technical failure, low infection), and no statistical outliers were removed. All data points were included in the analysis. The number of biological replicates was chosen based on common standards in the field and consistency across experiments. Measures of central tendency and variability are indicated in all graphs, and the statistical significance threshold was set at $p \leq 0.05$.

## Data availability

The mass spectrometry proteomics data have been deposited to the ProteomeXchange Consortium via the Pride partner repository (https://www.ebi.ac.uk/pride/) with the dataset identifiers PXD057162 for cell lysates (https://www.ebi.ac.uk/pride/archive/projects/PXD057162) and PXD057129 for viral particles (https://www.ebi.ac.uk/pride/archive/projects/PXD057129).

The source data of this paper are collected in the following database record: biostudies:S-SCDT-10_1038-S44319-025-00607-1.

## Peer review information

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

## Acknowledgements

We thank Stéphane Emiliani for helpful discussions and for critical reading of the manuscript. We thank all the members of the "Interactions hôte-virus" Laboratory for comments and helpful discussions. We thank the Imaging photonic Facility IMAG'IC and the Cytometry and Immunobiology Facility CYBIO of the Cochin Institute for technical assistance. We gratefully thank Dr Lazarou for the gift of the following cell lines: HeLa WT, HeLa LC3 TKO, GABARAP TKO (Nguyen et al, 2021). The following reagents were obtained through the National Institute for Biological Standards and Control Centralized Facility for AIDS Reagents, which is supported by the EU Program EVA and the UK Medical Research Council: HIV-1 gp120 monoclonal antibody (2G12) from Dr H. Katinger and mouse antibodies against CAp24 (EVA365; EVA366) from B. Wahren. The following reagents were obtained through the AIDS research and reference reagent program, Division of AIDS, NIAID, NIH: HIV-1 NL4-3 Vpu antiserum (NIH; ARP-969; from Drs K. Strebel and F. Maldarelli), rabbit monoclonal anti-MAp17 HIV-1 (NIH; ARP-4811; from Dr P. Spearman and Dr L. Ding), human polyclonal anti-HIV-1 SF2 p24 (NIH; ARP-4250), pNL4-3 ΔNef (ARP-12755; from Dr Olivier Schwartz), P4 MAGI CCR5+ cells (ARP-3580, from Dr N. Landau). We thank Dr K. Strebel for the gift of pNL4-3/Udel proviral DNA and Dr P. Bieniasz for the gift of pNL4-3 (MA/YFP) proviral DNA. DJ holds a fellowship from ANRS (ECTZ60924) and then from SIDACTION (2021-2-FJC-13113), CA from ANRS, MP from the "Ministère Français de l'enseignement supérieur et de la Recherche" and then from FRM (FDT202304016406). This work is funded by ANRS (ECTZ 158570 and ECTZ243795).

## Author contributions

**Marjory Palaric**: Conceptualization; Data curation; Formal analysis; Validation; Investigation; Visualization; Methodology; Writing—review and editing. **Margaux Versapuech**: Conceptualization; Data curation; Formal analysis; Investigation; Methodology; Writing—review and editing. **Delphine Judith**: Conceptualization; Data curation; Formal analysis; Investigation; Visualization; Methodology; Writing—review and editing. **Corentin Aubé**: Resources; Methodology. **Marjorie Leduc**: Resources; Data curation; Formal analysis; Investigation; Methodology; Writing—review and editing. **Jacques Dutrieux**: Methodology. **Emilie-Fleur Gautier**: Resources; Validation; Methodology. **Jean-Christophe Paillart**: Resources; Validation; Methodology; Writing—review and editing. **Sarah Gallois-Montbrun**: Conceptualization; Resources; Formal analysis; Validation; Visualization; Methodology; Writing—review and editing. **Clarisse Berlioz-Torrent**: Conceptualization; Data curation; Formal analysis; Supervision; Funding acquisition; Validation; Investigation; Visualization; Methodology; Writing—original draft; Project administration; Writing—review and editing.

Source data underlying figure panels in this paper may have individual authorship assigned. Where available, figure panel/source data authorship is listed in the following database record: biostudies:S-SCDT-10_1038-S44319-025-00607-1.

## Disclosure and competing interests statement

The authors declare no competing interests.

# Expanded View Figures

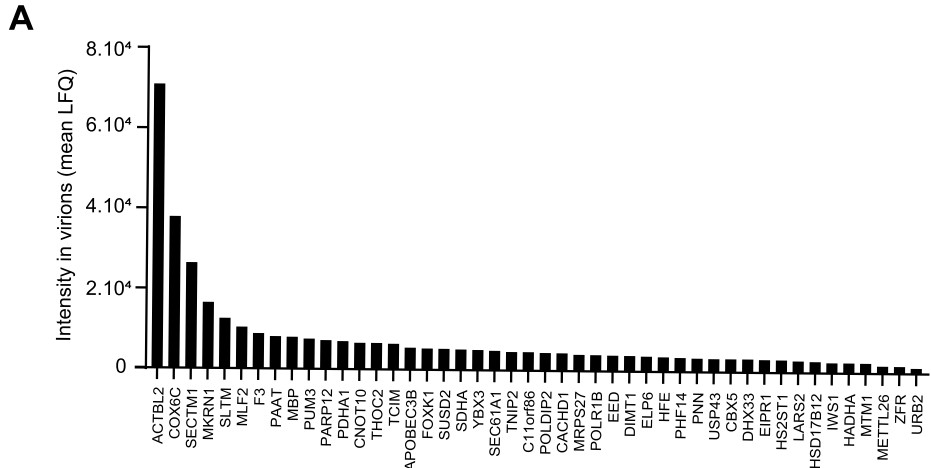

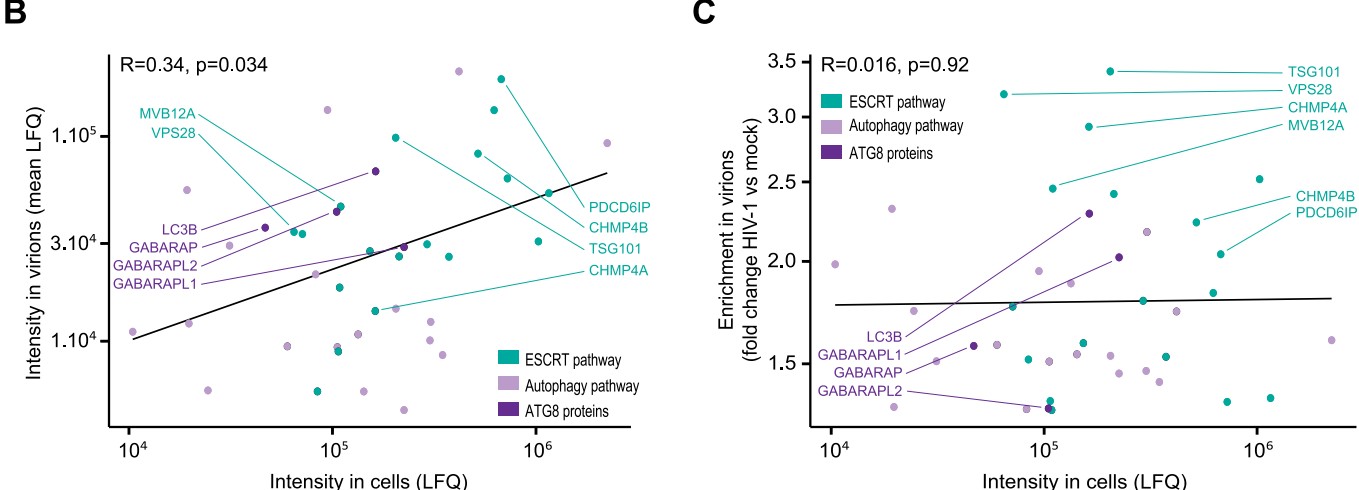

**Figure EV1.   Proteomic analysis of HIV-1 preparation and total cell lysate.**

HeLa cells were transfected with WT NL4-3 HIV-1 proviral DNA or non-transfected as a control for 48 h. Supernatants were collected, filtered and ultracentrifuged on a 20% sucrose cushion. Virions-enriched preparations were subsequently lysed and analyzed by LC-MS. (A) LC-MS signal intensities (LFQ) of cellular proteins identified exclusively in four viral preparations from NL4-3 HIV-1 transfected HeLa cells ($n = 4$ biological replicates). (B) LC-MS signal intensities (LFQ) of viral preparation-associated ATG-related proteins from four experiments were plotted against signal intensities of these proteins in the total cell lysate from one replicate. (C) Enrichment of viral preparation-associated ATG-related proteins compared to control preparation from mock-transfected cells, from four biological replicates, plotted against signal intensities of these proteins in the total cell lysate from one replicate. Spearman correlations were calculated using the GG Scatter Package from the GG PubR library (version 0.6.0.999) in R. Data information: Statistical analysis was performed using a one-tailed paired Student $t$-test. Source data are available online for this figure.

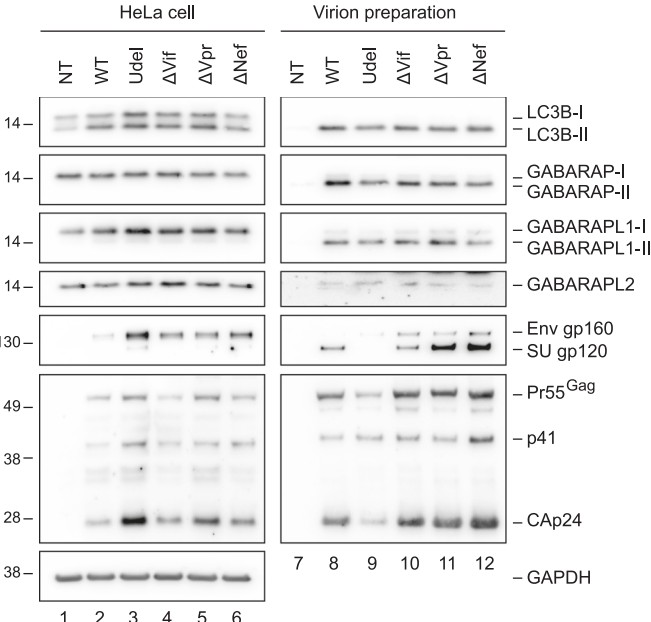

**Figure EV2. Effect of the absence of HIV-1 accessory gene expression on the incorporation of LC3/GABARAP proteins into viral particles.**

HeLa cells were transfected or not (NT) with WT, Udel, ΔVif, ΔVpr, and ΔNef NL4-3 HIV-1 proviral DNA for 48 h. Supernatants were collected and ultracentrifuged on a 20% sucrose cushion. Western blot analysis of LC3B, GABARAP, GABARAPL1, GABARAL2, HIV-1 SUgp120, HIV-1 Gag, and CAp24 products and GAPDH in producing cells and virion preparations. LC3B-I, GABARAP-I, and GABARAPL1-I correspond to the non-lipidated form of LC3B, GABARAP, and GABARAPL1, and LC3B-II, GABARAP-II, and GABARAPL1-II to the lipidated forms. Source data are available online for this figure.

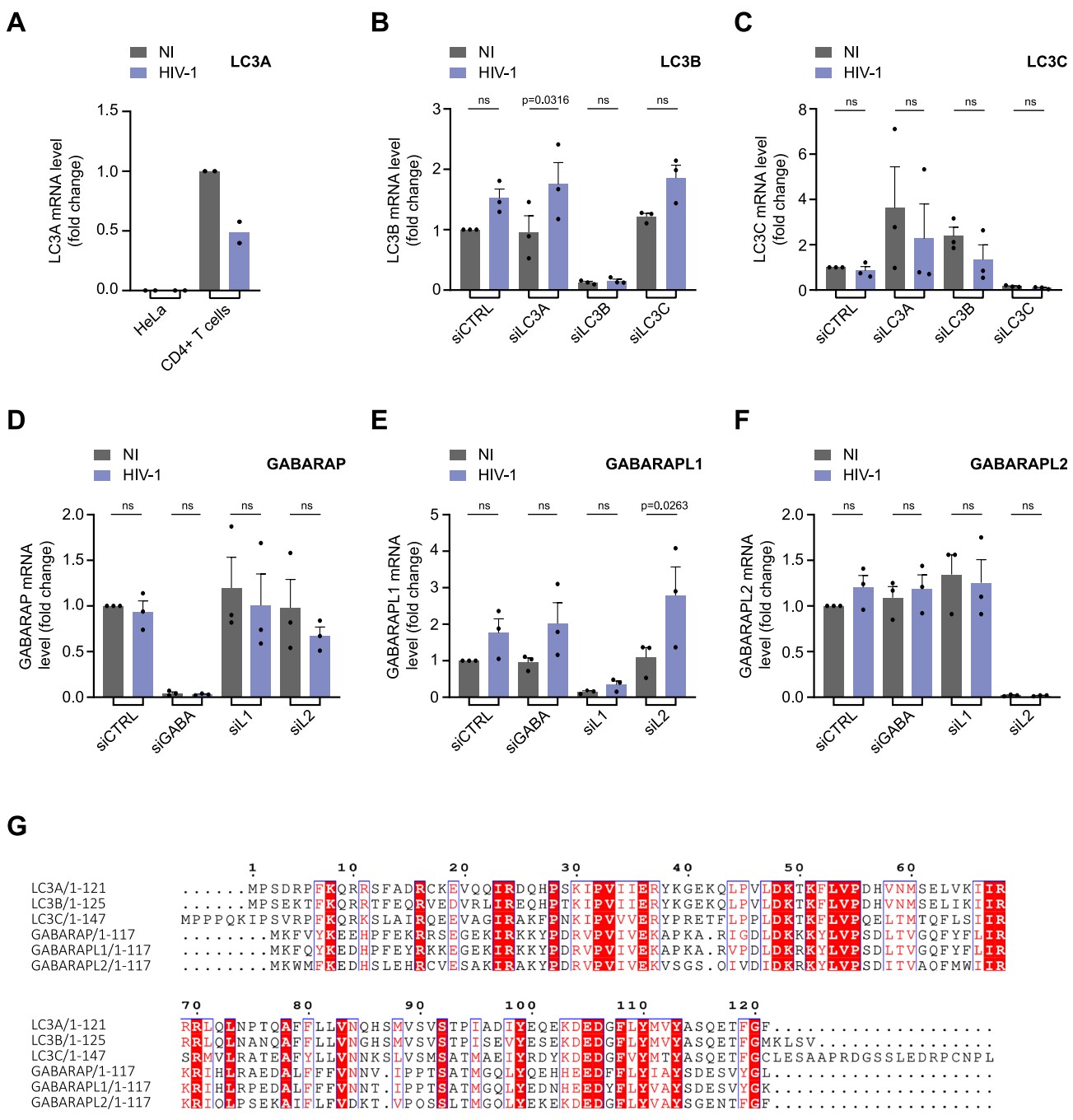

**Figure EV3.** **Quantification of LC3/GABARAP mRNA expression in HIV-1-infected cells.**

The mRNA quantity of LC3A was determined by RT-qPCR of total RNA extracted from HeLa cells or CD4 + T cells. HeLa cells or CD4 + T cells were infected or not (NI) with a VSVg pseudotyped WT NL4-3 HIV-1 at a MOI of 0.5 for 48 h. The mRNA quantity of LC3A was determined by RT-qPCR of total RNA extracted from cells. $n = 2$ biological replicates (**A**). HeLa cells transfected with control siRNA (siCTRL) or siRNA targeting LC3A, LC3B, LC3C, GABARAP, GABARAPL1, or GABARAPL2 were infected or not (NI) with a VSVg pseudotyped WT NL4-3 HIV-1 at a MOI of 0.5 for 48 h. The mRNA quantity of LC3B (**B**), LC3C (**C**), GABARAP (**D**), GABARAPL1 (**E**), or GABARAPL2 (**F**) was determined by RT-qPCR of total RNA extracted from cells. (**G**) Sequence alignment of LC3/GABARAP proteins using ESPript61. Identical residues are in red and similar ones are boxed. (**H**) Percentage of homology between LC3/GABARAP proteins. Data information: in (**B–F**), statistical analysis using a one-way ANOVA with Sidak's multiple comparisons test; mean ± SEM; $n = 3$ biological replicates; ns not significant. Source data are available online for this figure.

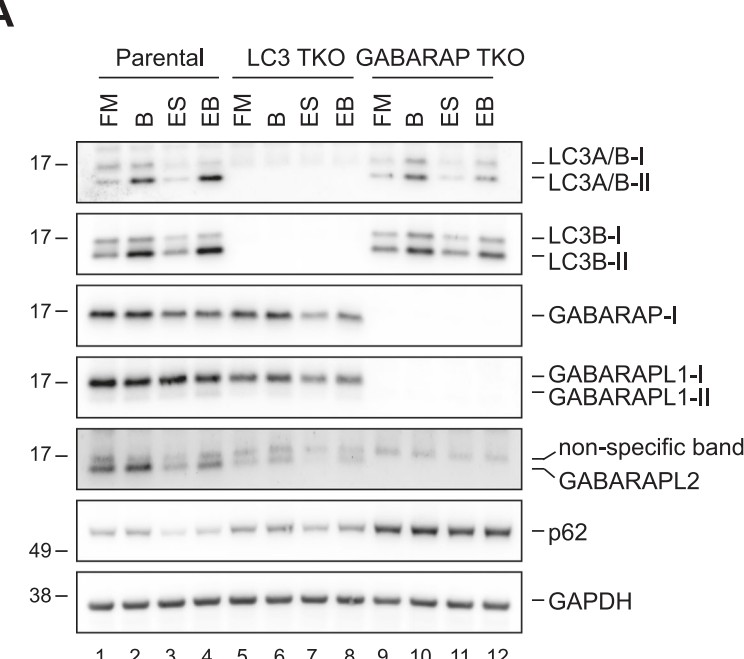

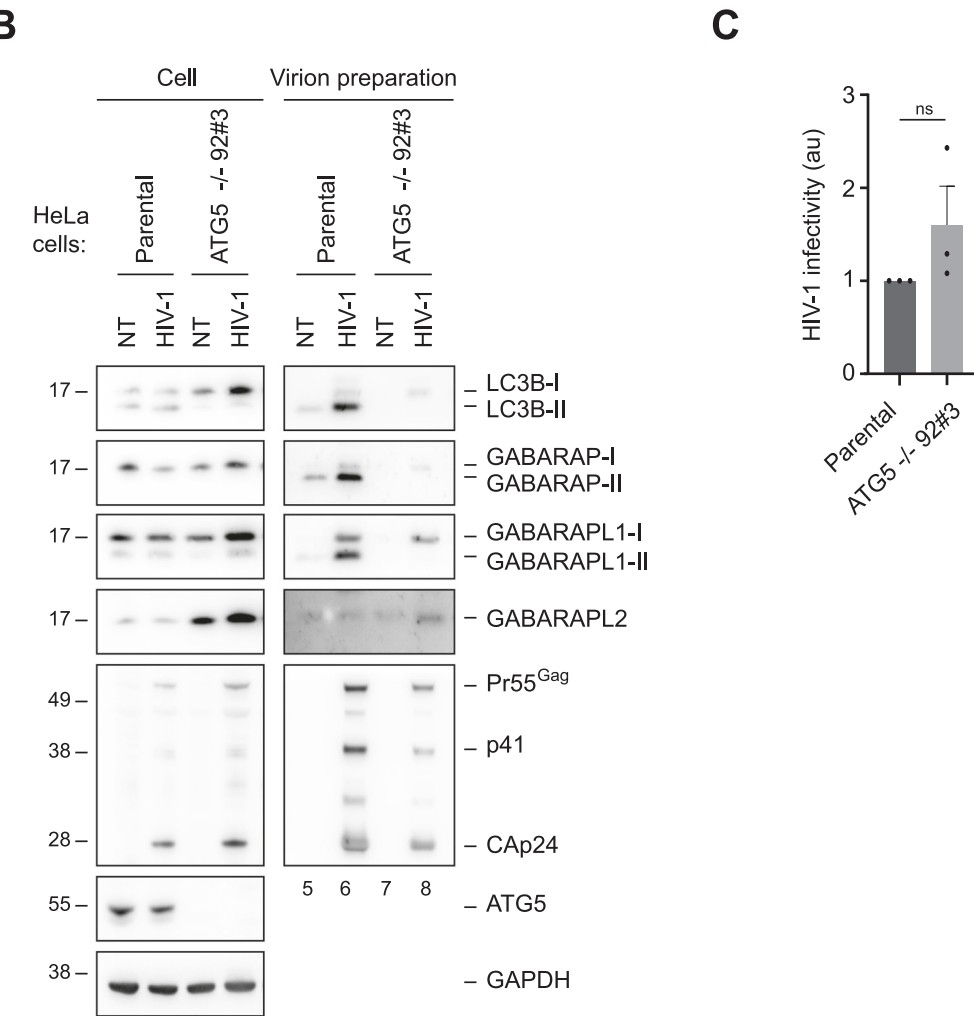

◀ **Figure EV4. Impact of LC3 and GABARAP knockout on autophagy flux in HeLa cells.**

(A) Parental, LC3 (LC3 TKO), or GABARAP (GABARAP TKO) knock down cells were incubated in full medium (FM), full medium with Bafilomycin A1 (B), EBSS for amino acid depletion without or with Bafilomycin A1 (ES and EB, respectively) for 2 h. Western blot analysis of LC3A/B, LC3B, GABARAP, GABARAPL1, GABARAPL2, and GAPDH in cells. LC3A/B-I, GABARAP-I, and GABARAPL1-I correspond to the non-lipidated form of LC3A/B, GABARAP and GABARAPL1, and LC3A/B-II, GABARAP-II and GABARAPL1-II to the lipidated forms. All western blots are representative of at least three biological replicates. (B) Parental or ATG5−/− 92#3 knockout HeLa cells were transfected or not (NT) with WT NL4-3 HIV-1 proviral DNA for 48 h. Cell lysates were recovered, and supernatants were collected, filtered, and ultracentrifuged on a 20% sucrose cushion. Western blot analysis of LC3B, GABARAP, GABARAPL1, GABARAPL2, HIV-1 Gag and CAp24 products, ATG5 and GAPDH in producing cells and virion preparations. All western blots are representative of at least three biological replicates. (C) The infectivity of released virus was determined by a β-galactosidase reporter assay and normalized by the quantity of released CAp24. Data information: Statistical analysis using a one-tail unpaired Student's *t*-test; mean ± SEM; *n* = 3 biological replicates; ns not significant. Source data are available online for this figure.

## A

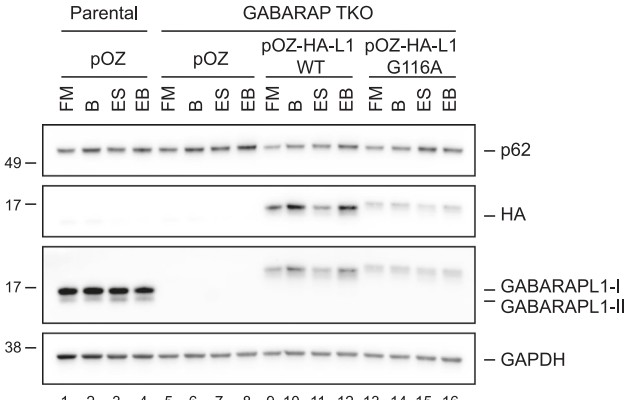

**Figure EV5. Transcomplementation of GABARAP TKO cells with wild-type and non-lipidated mutant forms of GABARAPL1.**

Parental, GABARAP (GABARAP TKO) knock down and HA-GABARAPL1 WT or G116A trans-complemented TKO cells were incubated in full medium (FM), full medium with Bafilomycin A1 (B), EBSS for amino acid depletion without or with Bafilomycin A1 (ES and EB, respectively) for 2 h. Western blot analysis of HA, GABARAPL1, p62, and GAPDH in cells. Source data are available online for this figure.

