## [Peer Review File · EMBO Reports]

GABARAP proteins regulate the packaging of HIV-1 genomic RNA into virions

Marjory Palaric, Margaux Versapuech, Delphine Judith, Corentin Aubé, Marjorie Leduc, Jacques Dutrieux, Emilie-Fleur Gautier, Jean-Christophe Paillart, Sarah Gallois-Montbrun, and Clarisse Berlioz-Torrent

Corresponding author(s): Clarisse Berlioz-Torrent (clarisse.berlioz@inserm.fr)

Review Timeline:

Submission Date:	24th Jan 25
Editorial Decision:	25th Mar 25
Revision Received:	25th Jul 25
Editorial Decision:	21st Aug 25
Revision Received:	24th Sep 25
Accepted:	9th Oct 25

Editor: Martina Rembold / Achim Breiling

Transaction Report:

Dear Dr. Berlioz-Torrent

Thank you for the submission of your research manuscript to our journal. I apologize for the delay in handling your manuscript but we have now received the full set of referee reports that is copied below.

As you will see, the referees acknowledge that the findings are interesting but they also raise a number of concerns and have suggestions how to further strengthen the data that all appear pertinent and need to be addressed.

Please let me know in case you disagree, and we can discuss the exact revision requirements further, also in a video chat, if you like.

Given these constructive comments, we would like to invite you to revise your manuscript with the understanding that the referee concerns (as detailed above and in their reports) must be fully addressed and their suggestions taken on board. Please address all referee concerns in a complete point-by-point response. Acceptance of the manuscript will depend on a positive outcome of a second round of review. It is EMBO Reports policy to allow a single round of revision only and acceptance or rejection of the manuscript will therefore depend on the completeness of your responses included in the next, final version of the manuscript.

We realize that it is difficult to revise to a specific deadline. In the interest of protecting the conceptual advance provided by the work, we recommend a revision within 3 months (June 25). Please discuss the revision progress ahead of this time with the editor if you require more time to complete the revisions.

*****IMPORTANT NOTE:

We perform an initial quality control of all revised manuscripts before re-review. Your manuscript will FAIL this control and the handling will be delayed IN CASE the following APPLIES:

- 1) A data availability section providing access to data deposited in public databases is missing. If you have not deposited any data, please add a sentence to the data availability section that explains that.
- 2) Your manuscript contains statistics and error bars based on $n=2$. Please use scatter blots in these cases. No statistics should be calculated if $n=2$.

When submitting your revised manuscript, please carefully review the instructions that follow below. Failure to include requested items will delay the evaluation of your revision. *****

- 1) a .docx formatted version of the manuscript text (including legends for main figures, EV figures and tables). Please make sure that the changes are highlighted to be clearly visible.
- 2) individual production quality figure files as .eps, .tif, .jpg (one file per figure). Please download our Figure Preparation Guidelines (figure preparation pdf) from our Author Guidelines pages <https://www.embopress.org/page/journal/14693178/authorguide> for more info on how to prepare your figures.
- 3) a .docx formatted letter INCLUDING the reviewers' reports and your detailed point-by-point responses to their comments. As part of the EMBO Press transparent editorial process, the point-by-point response is part of the Review Process File (RPF), which will be published alongside your paper.
- 4) a complete author checklist, which you can download from our author guidelines (<<https://www.embopress.org/page/journal/14693178/authorguide>>). Please insert information in the checklist that is also reflected in the manuscript. The completed author checklist will also be part of the RPF.
- 5) Please note that all corresponding authors are required to supply an ORCID ID for their name upon submission of a revised manuscript (<<https://orcid.org/>>). Please find instructions on how to link your ORCID ID to your account in our manuscript tracking system in our Author guidelines (<<https://www.embopress.org/page/journal/14693178/authorguide#authorshipguidelines>>)
- 6) We replaced Supplementary Information with Expanded View (EV) Figures and Tables that are collapsible/expandable online.

A maximum of 5 EV Figures can be typeset. EV Figures should be cited as "Figure EV1, Figure EV2" etc... in the text and their respective legends should be included in the main text after the legends of regular figures.

7) Before submitting your revision, primary datasets (and computer code, where appropriate) produced in this study need to be deposited in an appropriate public database (see <<https://www.embopress.org/page/journal/14693178/authorguide#dataavailability>>).

The accession numbers and database should be listed in a formal "Data Availability" section (placed after Materials & Method) that follows the model below (see also <<https://www.embopress.org/page/journal/14693178/authorguide#dataavailability>>). Please note that the Data Availability Section is restricted to new primary data that are part of this study.

Data availability

Additional information on source data and instruction on how to label the files are available <<https://www.embopress.org/page/journal/14693178/authorguide#sourcedata>>.

10) Figure legends and data quantification:

- the name of the statistical test used to generate error bars and P values,
 - the EXACT p-values,
 - the number (n) of independent experiments (please specify technical or biological replicates) underlying each data point,
 - the nature of the bars and error bars (s.d., s.e.m.)
- If the data are obtained from n {less than or equal to} 5, show the individual data points in addition to the SD or SEM.
- If the data are obtained from n {less than or equal to} 2, use scatter blots showing the individual data points.

11) Our journal encourages inclusion of *data citations in the reference list* to directly cite datasets that were re-used and obtained from public databases. Data citations in the article text are distinct from normal bibliographical citations and should directly link to the database records from which the data can be accessed. In the main text, data citations are formatted as

follows: "Data ref: Smith et al, 2001" or "Data ref: NCBI Sequence Read Archive PRJNA342805, 2017". In the Reference list, data citations must be labeled with "[DATASET]". A data reference must provide the database name, accession number/identifiers and a resolvable link to the landing page from which the data can be accessed at the end of the reference. Further instructions are available at <<https://www.embopress.org/page/journal/14693178/authorguide#referencesformat>>.

12) All Materials and Methods need to be described in the main text using our 'Structured Methods' format. According to this format, the Methods section includes a Reagents and Tools Table (listing key reagents, experimental models, software and relevant equipment and including their sources and relevant identifiers) followed by a Methods and Protocols section describing the methods, ideally using a step-by-step protocol format. The aim is to facilitate adoption of the methodologies across labs. Please download and fill our Reagents and Tools Table template (.docx), which you can find in our author guidelines: <https://www.embopress.org/page/journal/14693178/authorguide#structuredmethods>.

13) As part of the EMBO publication's Transparent Editorial Process, EMBO Reports publishes online a Review Process File to accompany accepted manuscripts. This File will be published in conjunction with your paper and will include the referee reports, your point-by-point response and all pertinent correspondence relating to the manuscript.

Yours sincerely,

=====

Referee #1:

This manuscript reports on the effects of GABARAP in the formation of HIV-1 virions and viral infectivity. GABARAP is a member of a family of proteins (known as ATG proteins) that control autophagy. Palaric et al. used proteomic analysis to identify proteins associated with HIV-1 in HeLa cells and among them, they focused on members of the LC3/GABARAP family. Their results show that LC3B, GABARAP, GABARAPL1 and GABARAPL2 were relevant for HIV propagation, showing evidence of the role of GABARAP in HIV genomic RNA packaging.

The study is rather comprehensive, logical in its design, and experiments appear to be technically well-done. However, there are several issues than in my opinion remain unclear and need revision.

1) The results of the proteomic analysis point towards members of the LC3/GABARAP family as factors affecting virion production and infectivity. Further validation of these observations suggest a role for LC3B, GABARAP, GABARAPL1 and GABARAPL2. However, this role is not the same for all of them (e.g., Figure 6B), and the effects of LC3B appear to be rather different from those of GABARAP proteins. The manuscript concentrates on GABARAP while LC3B appears to be left aside and there is little information or attempts to assign a function for LC3B.

The fact that authors do not comment a lot about LC3B function while describing extensively GABARAP effects is not clearly justified, particularly when the paper refers to LC3/GABARAP family as the same entity.

This needs to be more extensively discussed. It is also important that authors provide structural information (amino acid sequence alignments, and functional roles, major properties) of the six members of the ATG family in humans. What are the differences between the three GABARAP proteins (among them and with LC3B)? This would be a necessary item in supplementary materials to understand the rationale of the paper, and understand the conclusions of the study.

I checked with Uniprot some of this info, but it was not clear to me (<https://www.uniprot.org/uniprotkb/O95166/entry>)

2) The Discussion is too long and reads like a recapitulation of the results, while missing important points such as potential explanations for the difference between LC3B and GABARAP. In addition, Figures like Fig. 6 cannot be easily interpreted without having a clear idea of the protein composition. Authors need to explain what are GABARAPL-I and -II, in addition to the other members as indicated above.

3) One of the limitations of this study is the potential contamination with exosomes (as recognized by authors in the Discussion). I wonder if the 7SL RNA is an adequate control in these experiments. Does it have the same size as gHIV RNA. Some discussion is required here.

Other issues:

1) Introduction, lines 8-9: The sentence "Controlling the elongation..." is not complete and needs to be re-written.

2) Page 4, line 8: EBV, VZV, HCMV are common abbreviations for virologists but have to be defined.

3) Page 7, first line of the last paragraph: The 41 proteins are not mentioned earlier. They need to be defined in a clearer manner.

4) Page 8, first paragraph: The correlations mentioned are very poor, I think that authors should remove part of the text and tone down their statements.

5) Colors of the dots in panel C (Figure 1) are not distinguishable.

6) Figure 2C (NT stands for? May be, this is Mock-transfected?)

7) Legend to Figure 5. In the first line authors refer to LC3. But there is no LC3 TKO in the figure. Please re-phrase accordingly.

Referee #2:

In this manuscript, Palaric et al., through various assays, provide evidence that some ATG8 proteins, including LC3B, GABARAP, GABARAPL1, and GABARAPL2, which are essential components of the cellular recycling process, autophagy, are incorporated into purified HIV particles. Despite similar incorporation efficiency, they show that GABARAPs, not LC3B, are essential for infectious HIV particle production. Knockout of GABARAPs (GABARAP TKO), intriguingly, reduced HIV packaging and infectivity without impacting viral release.

While these experiments offer valuable insights into the roles of GABARAP proteins in HIV morphogenesis, further experimental evidence is needed to support the manuscript's main findings and enhance its scientific rigor and impact.

Comments:

The manuscript did not provide direct experimental evidence that GABARAP proteins regulate HIV assembly. An EM experiment, as well as measuring intra- and extracellular HIV titers in GABARAP TKO cells, is needed to confirm that these proteins regulate HIV packaging.

Figures 2C: No transfection (NT) control is shown in the figure. However, in the figure legend, the authors wrote mock-transfection. This typo should be corrected.

Figure 3A: The authors show that the knockdown of LC3C leads to the accumulation of cell-associated Cap24 and a reduction in Cap24 levels in viral preparations, indicating that LC3C is essential for efficient HIV release, as shown in Figure 1C. However, LC3 TKO, as shown in Figure 4A, did not increase cell-associated Cap24 but reduced viral release (Figure 4D). Similarly, the western blot data in Figure 4A indicates that GABARAP TKO increases HIV release (low cell-associated Cap24 vs high virion-associated Cap24). Yet, the viral release data in Figure 4D shows no significant effect of GABARAP TKO on HIV release. Could the authors explain this contradiction?

Figure 5C: The effect of GABARAP TKO on HIV packaging, which is one of the main findings of this manuscript, is marginal. It is unclear how a less than two-fold reduction in viral envelopment will translate in vivo.

Also, could the authors explain why GABARAP TKO attenuates viral RNA packaging (Figure 5C) but has no significant impact on viral release (Figure D)?

Figure 5C. Could the authors indicate in the materials and methods how released RNA measured by qPCR is normalized to released Cap24 measured by ELISA?

Figure 6B: The authors conclude that GABARAP-Gag interaction depends on RNA, given that RNase treatment attenuated GABARAP-Gag interaction. However, there appears to be less HA in the RNase-treated control (line 12) compared to no RNA treatment (line 11). Can the authors normalize Gag expression to HA to show that the reduction in GABARAP-Gag interaction is not a result of low expression of GABARAP in the RNase-treated group?

Figures 6C: The variation between the replicates in the UV-crosslinked GABARAPL1 group is too high, making it difficult to interpret the data.

Figure 6H. Similar to Figure 6C, it is challenging to believe that GABARAP TKO reduces HIV1 genomic RNA membrane association, given the large variations between replicates.

Supplemental Figure 4A. P62/SQSTM1 western blot is needed to confirm the effects of the TKOs on autophagic flux.

Referee #3:

Palaric et al., demonstrate conclusively that at least four of the human ATG8 like molecules are present in HIV-1 infected cell supernatants and provide strong evidence that these ATG8s are in fact incorporated into HIV-1 particles. These are either lipidated or a mix of lipidated and unlipidated ATG8s depending on the cell types that were infected. They then provide evidence that the three GABARAPs (GABARAP, GABARAPL1 and GABARAPL2) are collectively required for producing optimally infectious HIV-1 particles in some way that does not depend on virion release. They evaluate genomic RNA packaging into virions and compare it to a packaged cellular RNA to show that GABARAPs are required for efficient genomic RNA packaging, but not for cellular RNA packaging. They then show an interaction between GABARAPL1 and HIV-1 Gag that depends on the presence of viral RNA, and show that GABARAPL1 interacts with HIV-1 gRNA. The findings are novel and the paper is interesting. In general the experiments are performed well.

Major comments: I have one major comment which I believe the authors must address with a series of new experiments before the manuscript is suitable for publication. Comparing knock-out cells to their parental cell line is always susceptible to the criticism that something other than the intentionally targeted genes is responsible for the observed phenotypes, either due to off-target effects or to adaptation by the KO cell line. The GABARAP TKO cells must be therefore complemented with GABARAPL1 (and/or potentially the other two GABARAPs) before these experiments can be interpreted with confidence. This is also important to address another claim made in the paper that I am sceptical about. The authors use a ATG5 KO line to make the claim that lipidation is unnecessary for the effects of GABARAPs on HIV RNA packaging. Once again this effect might not in fact be due to ATG5 deficiency. However, the authors could express a C-terminal glycine to alanine GABARAPL1 mutant that cannot be lipidated in the GABARAP TKO cell line. If this complemented the phenotype to the same extent as wild-type GABARAPL1 that would prove their point. Alternatively if it did not complement the observed phenotypes to the same extent as wild-type, especially at physiological expression levels, the authors would need to modify their claim. The manuscript could be enhanced by generating mutants of GABARAPL1 that failed to interact with genomic RNA and (presumably) therefore failing to complement the phenotypes described.

Minor comments: Statistical analysis of multiple independent groups should not be performed with Student's t-Test - there are several instances, for example Figure 3C-F. This probably will not change any of the main conclusions but the authors will need to demonstrate this with more appropriate statistical tests.

The manuscript is generally well written though a more up-to-date account of ATG8 lipidation would be welcome. LAP and LANDO are now usually referred to as a type of CASM, (conjugation of ATG8s to Single Membranes) or VAIL (V-ATPase ATG16L1 Induced Lipidation) and recently an ATG16L1 independent ATG8 lipidation pathway STIL (Sphingomyelin TECPR1 Induced Lipidation) has been described.

Dr Clarisse Berlioz-Torrent

Head of Host-Virus Interactions laboratory
27, rue du faubourg Saint Jacques
75014 Paris
clarisse.berlioz@inserm.fr
Tel : 33 (0) 1 40516575

Paris, July 25th, 2025

EMBO Reports Tracking #EMBOR-2025-61223V1

Point-by-point response to the reviewers' comments

Reviewer 1

This manuscript reports on the effects of GABARAP in the formation of HIV-1 virions and viral infectivity. GABARAP is a member of a family of proteins (known as ATG proteins) that control autophagy. Palaric et al. used proteomic analysis to identify proteins associated with HIV-1 in HeLa cells and among them, they focused on members of the LC3/GABARAP family. Their results show that LC3B, GABARAP, GABARAPL1 and GABARAPL2 were relevant for HIV propagation, showing evidence of the role of GABARAP in HIV genomic RNA packaging. The study is rather comprehensive, logical in its design, and experiments appear to be technically well-done. However, there are several issues that in my opinion remain unclear and need revision.

1) The results of the proteomic analysis point towards members of the LC3/GABARAP family as factors affecting virion production and infectivity. Further validation of these observations suggest a role for LC3B, GABARAP, GABARAPL1 and GABARAPL2. However, this role is not the same for all of them (e.g., Figure 6B), and the effects of LC3B appear to be rather different from those of GABARAP proteins. The manuscript concentrates on GABARAP while LC3B appears to be left aside and there is little information or attempts to assign a function for LC3B.

The fact that authors do not comment a lot about LC3B function while describing extensively GABARAP effects is not clearly justified, particularly when the paper refers to LC3/GABARAP family as the same entity. This needs to be more extensively discussed.

The reviewer is right: the roles of LC3 and GABARAP proteins in the HIV-1 replication cycle are different. We have previously shown that LC3C is important for efficient virus production

(Madjo et al., *Cell Reports* 2016). Therefore, in this study, we deliberately focused on the GABARAP proteins and their previously undescribed functions in the late stages of the HIV-1 life cycle. For clarity, we have now clarified this point on page 15 of the Results section.

It is also important that authors provide structural information (amino acid sequence alignments, and functional roles, major properties) of the six members of the ATG family in humans. What are the differences between the three GABARAP proteins (among them and with LC3B)? This would be a necessary item in supplementary materials to understand the rationale of the paper, and understand the conclusions of the study. I checked with Uniprot some of this info, but it was not clear to me.

To improve clarity, additional information about LC3 and GABARAP proteins has been added to the Introduction section of our manuscript (page 3) and to the Results section (page 13). An additional figure showing the sequence homology between LC3/GABARAP proteins is now included as Figure EV3G-H.

2) The Discussion is too long and reads like a recapitulation of the results, while missing important points such as potential explanations for the difference between LC3B and GABARAP. In addition, Figures like Fig. 6 cannot be easily interpreted without having a clear idea of the protein composition.

We have rewritten and shortened the Discussion (pages 19-23) and added a paragraph addressing the functional differences between LC3 and GABARAP proteins in the viral replication cycle (page 23).

Authors need to explain what are GABARAP-I and -II, in addition to the other members as indicated above.

This information is now included in the Introduction (page 3), in the Results section of the manuscript (page 10), and in the legends of the western blot figures 2-4, 6-7, EV2, EV4.

3) One of the limitations of this study is the potential contamination with exosomes (as recognized by authors in the Discussion). I wonder if the 7SL RNA is an adequate control in these experiments. Does it have the same size as gHIV RNA. Some discussion is required here. Several studies have shown that 7SL RNA is specifically and abundantly incorporated into HIV-1 particles, rather than being randomly included or resulting from exosomal contamination (Itano et al., *Traffic* 2018; Eckwahl et al., *RNA* 2016; Didierlaurent et al., *Nucleic Acids Res.* 2011). Its selection is mediated by early interactions with Gag proteins, independently of

2

ESCRT pathway or genomic RNA packaging. Although 7SL is smaller (~300 nt) than the HIV-1 genomic RNA (~9 kb), it has been used as a stable internal reference for normalization of encapsidation in certain studies (e.g., K. GC, *J. Virol.* 2025; Didierlaurent et al., *Nucleic Acids Res.* 2011; Khan et al. *Retrovirology* 2007). Its well-characterized and reproducible incorporation profile makes it a reliable marker of virion RNA content rather than a byproduct of exosome contamination. Furthermore, to exclude exosome contamination, we immunoprecipitated HIV-1 particles using an anti-envelope antibody and found that gRNA levels were reduced more than two-fold in virions from GABARAP-deficient cells compared to control (Figure 5E–F), supporting a role for GABARAP proteins in HIV-1 genome packaging.

Other issues:

1) Introduction, lines 8-9: The sentence "Controlling the elongation..." is not complete and needs to be re-written.

The sentence has been corrected (page 3).

2) Page 4, line 8: EBV, VZV, HCMV are common abbreviations for virologists but have to be defined.

The full names of the viruses were provided on page 4.

3) Page 7, first line of the last paragraph: The 41 proteins are not mentioned earlier. They need to be defined in a clearer manner.

The 41 proteins have been defined page 8.

4) Page 8, first paragraph: The correlations mentioned are very poor, I think that authors should remove part of the text and tone down their statements.

We have rewritten the paragraph and town down our statements (Page 9)

5) Colors of the dots in panel C (Figure 1) are not distinguishable.

We have changed the color and the size of the dots in Figure 1C.

6) Figure 2C (NT stands for? May be, this is Mock-transfected?)

NT corresponds to non-transfected cells. This information has been included in all the figure legends.

7) Legend to Figure 5. In the first line authors refer to LC3. But there is no LC3 TKO in the figure. Please re-phrase accordingly.

The sentence has been rephrased to remove any mention of LC3 TKO cells.

#Reviewer 2

In this manuscript, Palaric et al., through various assays, provide evidence that some ATG8 proteins, including LC3B, GABARAP, GABARAPL1, and GABARAPL2, which are essential components of the cellular recycling process, autophagy, are incorporated into purified HIV particles. Despite similar incorporation efficiency, they show that GABARAPs, not LC3B, are essential for infectious HIV particle production. Knockout of GABARAPs (GABARAP TKO), intriguingly, reduced HIV packaging and infectivity without impacting viral release.

While these experiments offer valuable insights into the roles of GABARAP proteins in HIV morphogenesis, further experimental evidence is needed to support the manuscript's main findings and enhance its scientific rigor and impact.

The manuscript did not provide direct experimental evidence that GABARAP proteins regulate HIV assembly. An EM experiment, as well as measuring intra- and extracellular HIV titers in GABARAP TKO cells, is needed to confirm that these proteins regulate HIV packaging.

Reviewer 2 stated that “the manuscript did not provide direct experimental evidence that GABARAP proteins regulate HIV assembly”

HIV assembly corresponds to the step wherein the HIV particle is created at the plasma membrane and essential components (Env glycoprotein, accessory viral proteins, and viral genomic RNA) are packaged into HIV particle, which is formed by Gag and Gag-Pol viral precursors. In our study, we did quantify HIV-1 released in the supernatant (extracellular CAp24; Figure 4C) relative to the amount of CAp24 produced within cells (Figure 4B) and showed that GABARAP TKO does not modify the release/production of HIV-1 particles. However, GABARAP TKO affects viral genomic RNA packaging into viral particles, thus affecting HIV-1 assembly and infectivity.

This statement is supported by the fact that

- While the release of HIV particle in the supernatant remains unchanged in GABARAP TKO cells (Figure 4D), the infectivity of the released HIV particle is reduced (Figure 4E).

4

- This decrease does not result from a defect of HIV-1 Env glycoprotein incorporation into the particle (Figure 4A), but rather correlates with impaired packaging of the viral genomic RNA into the viral particles (Figure 5C-D).
- This RNA packaging defect is further confirmed in immunoprecipitated viral particles (Figure 5E-F).

Altogether, these experiments show that GABARAP proteins regulate RNA packaging and thus HIV-1 assembly.

Reviewer 2 requested the inclusion of an EM experiment to confirm the role of GABARAP proteins in RNA packaging, along with the measurements of intra- and extracellular HIV titers in GABARAP TKO cells

The nucleation of the core and the acquisition of the conical morphology can occur correctly even in the absence of HIV genomic RNA: the presence of genomic RNA and the condensation of the ribonucleoprotein complex are dispensable for the assembly of a mature capsid structure within the HIV-1 virion (Mattei et al., JVI 2015). Therefore, we believe that EM experiments are unlikely to effectively distinguish viral particles based on the presence or absence of packaged viral genomic RNA.

Intracellular HIV titers cannot be quantified as there are no infectious viruses inside the cell. HIV assembly occurs at the plasma membrane and leads to the budding of infectious viral particle outside the cell. The amount of HIV-1 released in the supernatant have been quantified by p24 ELISA as shown in figure 4C, and the infectivity of HIV-1 particles was determined by infection of the indicator cells HeLa P4R5 in a standardized 96-well titration assay by luminometric analysis of β -galactosidase activity, then normalized by the quantity of released CAp24 (Figure 4E).

Overall, we believe that EM and Intracellular titer will not yield additional relevant information (or are not doable) and we did not perform them.

To complement the data presented in Figure 6, we conducted RNAscope experiments to detect HIV-1 gRNA puncta in both parental and GABARAP TKO cells, infected with VSV-G-pseudotyped HIV-1 WT MA-YFP, a virus bearing a YFP fluorescent marker in Gag precursor. The results, shown in the new Figure 7C to 7E, reveal that in absence of GABARAPs proteins, the HIV-1 gRNAs accumulate within the cell, appearing less frequently at the plasma membrane. These findings support the data shown in the new Figures 7A–B, which demonstrate that viral genomic RNA is less associated with membrane fractions in GABARAP-deficient cells.

Altogether, these new results suggest that GABARAP may play a role in the molecular mechanism responsible for targeting HIV-1 genomic RNA to the assembly site at the plasma membrane. These data have been integrated into the revised manuscript: Figure 7C-E, Results section (page 19), Materials and Methods (pages 35-37), and Discussion (page 22).

Reviewer 2 also mentioned several additional points that we will answer as followed:

Figures 2C: No transfection (NT) control is shown in the figure. However, in the figure legend, the authors wrote mock-transfection. This typo should be corrected.

We have corrected it.

Figure 3A: The authors show that the knockdown of LC3C leads to the accumulation of cell-associated Cap24 and a reduction in Cap24 levels in viral preparations, indicating that LC3C is essential for efficient HIV release, as shown in Figure 1C. However, LC3 TKO, as shown in Figure 4A, did not increase cell-associated Cap24 but reduced viral release (Figure 4D). Similarly, the western blot data in Figure 4A indicates that GABARAP TKO increases HIV release (low cell-associated Cap24 vs high virion-associated Cap24). Yet, the viral release data in Figure 4D shows no significant effect of GABARAP TKO on HIV release. Could the authors explain this contradiction?

As correctly noted by the reviewer, we previously demonstrated that LC3C is essential for efficient HIV-1 particle release (Madjo, *Cell Reports*, 2016). The absence of LC3C leads to reduced viral particle release into the supernatant, often accompanied by an accumulation of cell-associated viral components when infection is efficient (Figure 3A and C).

In contrast, LC3 TKO cells exhibit unexplained lower infection or transfection efficiency compared to parental cells, resulting in reduced accumulation of cell-associated viral products. This reduction reflects decreased overall viral production rather than enhanced particle release. To account for this, we normalize the amount of virus in the supernatant to the intracellular production of viral components—a parameter we refer to as "HIV-1 release" (see Results, page 12 and Materials and Methods, page 29). Accordingly, HIV-1 release appears reduced in LC3 TKO cells due to the lower relative amount of extracellular virus normalized to intracellular levels.

Regarding GABARAP TKO cells, contrary to the reviewer's understanding, GABARAP deficiency does not alter HIV-1 release. While we observe increased levels of viral particles in the supernatant (Figure 4A and C), this mirrors a parallel increase in intracellular viral production

(Figure 4B). Consequently, when normalized, HIV-1 release remains unchanged compared to control cells (Figure 4D).

Figure 5C: The effect of GABARAP TKO on HIV packaging, which is one of the main findings of this manuscript, is marginal. It is unclear how a less than two-fold reduction in viral envelopment will translate *in vivo*.

The impact of GABARAP TKO on HIV RNA packaging is significant (Figure 5D) and even more pronounced when RNA packaging is assessed in immunopurified HIV particles (showing a reduction of over 60%) (Figure 5E). We could reasonably expect that if a substantial proportion of viral particles lack genomic RNA, viral replication would be impaired. Notably, the RNA packaging defect observed in the absence of GABARAP protein expression is comparable to the defects previously reported in the absence of the dimerization and packaging signals on the viral genomic RNA, which led to 50% to 80% decrease in RNA packaging (Paillart et al. J. Virol 1996; Lauhgrea et al. J. Virol 1997; Houzet et al. NAR 2007). In these latter studies, the impairment of genomic RNA packaging led to a strongly reduced viral dissemination (>200-fold).

Also, could the authors explain why GABARAP TKO attenuates viral RNA packaging (Figure 5C) but has no significant impact on viral release (Figure D)?

As already mentioned, viral genomic RNA is not required for the assembly, budding and envelopment of the HIV-1 capsid. HIV-1 particles can be produced even in the absence of viral genomic RNA packaging. These viral particles lacking genomic RNA, while present in the extracellular medium, are non-infectious and exhibit EM ultrastructural features similar to those of infectious viral particles that do contain viral genomic RNA.

Figure 5C. Could the authors indicate in the materials and methods how released RNA measured by qPCR is normalized to released CAP24 measured by ELISA?

We will mention it in material and methods (page 34).

Figure 6B: The authors conclude that GABARAP-Gag interaction depends on RNA, given that RNase treatment attenuated GABARAP-Gag interaction. However, there appears to be less HA in the RNase-treated control (line 12) compared to no RNA treatment (line 11). Can the authors normalize Gag expression to HA to show that the reduction in GABARAP-Gag interaction is not a result of low expression of GABARAP in the RNase-treated group?

We have normalized Gag to HA-GABARAPL1 expression found in our immunoprecipitation and introduce these data in this article (Figure 6C). The quantifications confirm that the RNase treatment attenuated GABABARP-Gag interaction as indicated in our result section (pages 16-17).

Figures 6C: The variation between the replicates in the UV-crosslinked GABARAPL1 group is too high, making it difficult to interpret the data.

This experiment has been repeated 2 more times to reduce the variation. The new assays have been introduced in figure 6D and enhanced the significance of our results. Altogether, this set of experiments reinforce our observations and conclusions.

Figure 6H. Similar to Figure 6C, it is challenging to believe that GABARAP TKO reduces HIV1 genomic RNA membrane association, given the large variations between replicates. This experiment has been repeated 3 more times to reduce the variation. The new assays have been introduced in the new figure 7B and enhanced the significance of our results. Altogether, this set of experiments reinforce our observations and conclusions.

Supplemental Figure 4A. P62/SQSTM1 western blot is needed to confirm the effects of the TKOs on autophagic flux.

We have added p62/SQSTM1 Western Blot in new figure EV4.

Reviewer 3

Palaric et al., demonstrate conclusively that at least four of the human ATG8 like molecules are present in HIV-1 infected cell supernatants and provide strong evidence that these ATG8s are in fact incorporated into HIV-1 particles. These are either lipidated or a mix of lipidated and unlipidated ATG8s depending on the cell types that were infected. They then provide evidence that the three GABARAPs (GABARAP, GABARAPL1 and GABARAPL2) are collectively required for producing optimally infectious HIV-1 particles in some way that does not depend on virion release. They evaluate genomic RNA packaging into virions and compare it to a packaged cellular RNA to show that GABARAPs are required for efficient genomic RNA packaging, but not for cellular RNA packaging. They then show an interaction between GABARAPL1 and HIV-1 Gag that depends on the presence of viral RNA, and show that GABARAPL1 interacts with HIV-1 gRNA. The findings are novel and the paper is interesting. In general the experiments are performed well.

8

Major comments: I have one major comment which I believe the authors must address with a series of new experiments before the manuscript is suitable for publication.

Reviewer 3 has a major comment.

He/she stated that "Comparing knock-out cells to their parental cell line is always susceptible to the criticism that something other than the intentionally targeted genes is responsible for the observed phenotypes, either due to off-target effects or to adaptation by the KO cell line. The GABARAP TKO cells must be therefore complemented with GABARAPL1 (and/or potentially the other two GABARAPs) before these experiments can be interpreted with confidence. This is also important to address another claim made in the paper that I am skeptical about. The authors use a ATG5 KO line to make the claim that lipidation is unnecessary for the effects of GABARAPs on HIV RNA packaging. Once again, this effect might not in fact be due to ATG5 deficiency. However, the authors could express a C-terminal glycine to alanine GABARAPL1 mutant that cannot be lipidated in the GABARAP TKO cell line. If this complemented the phenotype to the same extent as wild-type GABARAPL1 that would prove their point. Alternatively, if it did not complement the observed phenotypes to the same extent as wild-type, especially at physiological expression levels, the authors would need to modify their claim.

As requested by the reviewer, we performed transcomplementation assays. Stable GABARAP triple knockout (TKO) cell lines re-expressing GABARAPL1 protein or its unlipidated mutated variant (GABARAPL1 G116A) were generated by retroviral vector transduction (Figure EV5). HIV-1 RNA release and packaging were evaluated 48 hours post-infection of these cell lines. The results showed that the transcomplementation with GABARAPL1 protein or its lipidated form partially restore the release and the packaging of HIV-1 genomic RNA (Figure 5G-I). These findings support our previous results and conclusions, indicating that GABARAP proteins are involved in HIV-1 gRNA packaging. Furthermore, this novel role appears to be independent of their lipidation. The new results were introduced page 16, and in figure 5G-I. The "material and methods" section was completed page 30.

The manuscript could be enhanced by generating mutants of GABARAPL1 that failed to interact with genomic RNA and (presumably) therefore failing to complement the phenotypes described.

Regarding this last point, we agree with reviewer 3 that GABARAPL1 mutants unable to interact with genomic RNA would be highly informative in confirming the observed

phenotype. However, due to the lack of available data in the literature on the RNA-binding domains of ATG8 molecules (Hwang et al. Nat Comm 2022), addressing this request would necessitate an extensive structure-function study of the GABARAPL1:viral genomic RNA interaction which we believe, goes beyond the scope of the current study and will, in fact, constitute the foundation of our future research work.

Minor comments: Statistical analysis of multiple independent groups should not be performed with Student's t-Test - there are several instances, for example Figure 3C-F. This probably will not change any of the main conclusions but the authors will need to demonstrate this with more appropriate statistical tests.

We performed more appropriate statistical analyses. These new analyses have been done on results presented in Figures 3, 4, 5, 6 and 7, and do not change our conclusions.

The manuscript is generally well written though a more up-to-date account of ATG8 lipidation would be welcome. LAP and LANDO are now usually referred to as a type of CASM, (conjugation of ATG8s to Single Membranes) or VAIL (V-ATPase ATG16L1 Induced Lipidation) and recently an ATG16L1 independent ATG8 lipidation pathway STIL (Sphingomyelin TECPR1 Induced Lipidation) has been described.

We introduced the notion of CASM in the introduction and discussion sections (pages 3-4) to update recent bibliographic developments related to ATG8 lipidation.

Dear Dr. Berlioz-Torrent,

Thank you for the submission of your revised manuscript to our editorial offices. We have now received the reports from the two referees that were asked to re-evaluate the study, you will find below. As you will see, both referees now support publication of the study in EMBO reports. Referee #3 has a further suggestion to improve the manuscript, I ask you to address in a final revised manuscript. Please also provide a final p-b-p-response to these points and my editorial requests below.

Editorial requests:

- I would suggest this title (without the 'The'):

GABARAP proteins regulate the packaging of HIV-1 genomic RNA into virions

- Please order the manuscript sections like this, using (only) these names:

Title page - Abstract - Keywords - Introduction - Results - Discussion - Methods - Data availability section - Acknowledgements - Disclosure and Competing Interests Statement - References - Figure legends - Expanded View Figure legends.

- Please provide the abstract (not more than 175 words) written in present tense throughout.

- Please name the section 'Data Collection' 'Data Availability' and remove now the access information for referees. Please make sure the datasets are public latest on the day of online publication of the study.

- We updated our journal's competing interests policy in January 2022 and request authors to consider both actual and perceived competing interests. Please review the policy <https://www.embopress.org/competing-interests> and update your competing interests if necessary. Please name this section 'Disclosure and Competing Interests Statement' and put it after the Acknowledgements section.

- The nomenclature of the EV figure legends in the manuscript text file is not correct: It should be 'Figure EVx' instead of 'Expanded View Figure X'.

- Please use our reference format (last name, first name initial; et al should be used after the 10th name; DOIs should only be used for preprints and datasets that have not been published yet):

- Per journal policy, we do not allow 'data not shown', which is stated in the manuscript (page 19). All data referred to in the paper should be displayed in the main or Expanded View figures, or an Appendix. Thus, please add these data (or change the text accordingly if these data are not central to the study). See:

<https://www.embopress.org/page/journal/14693178/authorguide#unpublisheddata>

- Please also make sure that each figure panel (main and EV figures) is called out separately and sequentially. Presently, there seem to be no separate callouts for panels 3F and 3G. Please check.

- Please check again that the number "n" for how many independent experiments were performed, their nature (biological versus technical replicates), the bars and error bars (e.g. SEM, SD) and the test used to calculate p-values is indicated in the respective figure legends. Please also check that all the p-values are explained in the legend, that exact p-values are listed and that these fit to those shown in the figure. Please provide statistical testing where applicable. Please avoid the phrase 'independent experiment' but clearly state if these were biological or technical replicates. Please also indicate (e.g. with n.s.) if testing was performed, but the differences are not significant. In case n=2, please show the data as separate datapoints without error bars and statistics. See also:

<http://www.embopress.org/page/journal/14693178/authorguide#statisticalanalysis>

If n<5, please show single datapoints for diagrams. Moreover:

- Please note that the exact p values are not provided in the legends of figures 2B, 4E, 5E, 6F, 7D, E.

- Please indicate the statistical test used for data analysis in the legends of figures 1C, D; 2A, B.

- Please add to each legend (main, EV figures and Appendix Figures, where applicable) a 'Data Information' section explaining the statistics used or providing information regarding replicates and scales. See:

- Please make sure that all the funding information is also entered into the online submission system and that it is complete and similar to the one in the acknowledgement section of the manuscript text file. Presently, the grant numbers entered in the system (ECTZ158570, FDT202304016406, ECTZ60924, 2021-2-FJC-13113) are missing in the manuscript text file; the grant

number 23515 mentioned in the Acknowledgements is missing in the submission system. Please check.

- Please confirm that for all Western blot panels in the manuscript the loading control was run on the same gel as the other proteins detected. Please note that we discourage comparisons between samples on different gels/blots, even if the samples derive from one experiment, as confounding factors reduce comparability. If unavoidable, the figure legend must state that the samples derive from the same experiment and that gels/blots were processed in parallel. If a 'representative' loading control is shown for multiple gels/blots, the intra-gel controls should be shown in the source data files and the figure legends should describe the data displayed accurately. See our author guidelines:

<https://www.embopress.org/page/journal/14693178/authorguide#datapresentationformat> (section 'Electrophoretic gels and blots').

and

<https://www.embopress.org/image-integrity>

- Please add the primer information (Table EV2) directly to the Reagents & Tools table. Please add callouts to the Reagents & Tools table where appropriate. Please also remove Table EV2 from the manuscript files.

- Please add the legend for Table EV1 on the first TAB of the excel sheet. Then remove the legends for EV tables from the main manuscript text file.

- Thanks for providing the requested source data. However, please upload the source data as one folder per main figure, grouping together all the files (one for each panel) for this figure ZIPed together, and as one folder for the EV figures, grouping together all the files (one folder for each figure containing one file for each panel) ZIPed together.

In addition, I would need from you uploaded separately:

Best,

Referee #2:

This is a much-improved manuscript. The authors have adequately addressed/clarified all my concerns. I have no further comments.

Referee #3:

The authors have addressed my concerns (and that of other reviewers) with further experimental data. The key point I made was that complementation of the GABARAP triple knockout line would be necessary. The experimental data are presented in 5G-I. These are interpreted as showing partial complementation, but there are wide error bars. It looks like complementation was observed in 2/3 repeats. Once again an inappropriate statistical test has been used (although the authors corrected this elsewhere). My recommendation would be for the authors to repeat these to be sure. Otherwise I think the manuscript is suitable for publication.

Referee #3 request:

The authors have addressed my concerns (and that of other reviewers) with further experimental data. The key point I made was that complementation of the GABARAP triple knockout line would be necessary. The experimental data are presented in 5G-I. These are interpreted as showing partial complementation, but there are wide error bars. It looks like complementation was observed in 2/3 repeats. Once again an inappropriate statistical test has been used (although the authors corrected this elsewhere). My recommendation would be for the authors to repeat these to be sure. Otherwise I think the manuscript is suitable for publication.

We thank Reviewer 3 for the positive evaluation of our revised manuscript and for emphasizing the importance of the complementation experiment in GABARAP triple-knockout (TKO) cells.

In response, we repeated the complementation experiment three additional times, for a total of six independent biological replicates. The expanded dataset is presented in updated Figure 5H-I. As before, we assessed the ability of wild-type GABARAPL1 and a non-lipidated mutant (G116A) to restore HIV-1 genomic RNA (gRNA) packaging in GABARAP TKO cells.

To address the reviewer's concern regarding statistical analysis, we applied a Kruskal-Wallis test followed by Dunn's multiple comparison test, appropriate for non-parametric multi-group comparisons. Adjusted p-values are reported in the figure. This analysis confirms that trans-complementation with either wild-type or non-lipidated GABARAPL1 significantly restores HIV-1 gRNA packaging in virions produced from TKO cells (see updated figure panels and legends for details). Together, these results support our conclusion that GABARAPL1 rescues the gRNA-packaging defect in GABARAP TKO cells and that this rescue is lipidation-independent.

We hope these additional data and the corrected statistical approach fully address the reviewer's concern. We are grateful for this constructive suggestion, which has strengthened the manuscript.

Editorial requests:

- I would suggest this title (without the 'The'):

GABARAP proteins regulate the packaging of HIV-1 genomic RNA into virions

We have modified the title according to the recommendation.

- Please order the manuscript sections like this, using (only) these names:

Title page - Abstract - Keywords - Introduction - Results - Discussion - Methods - Data availability section - Acknowledgements - Disclosure and Competing Interests Statement - References - Figure legends - Expanded View Figure legends.

We have reorganized the manuscript as requested.

- Please provide the abstract (not more than 175 words) written in present tense throughout.

We have rewritten the abstract in present tense.

- Please name the section 'Data Collection' 'Data Availability' and remove now the access information for referees. Please make sure the datasets are public latest on the day of online publication of the study.

We have renamed the section 'Data Collection' 'Data Availability' and removed the access for referees

- We updated our journal's competing interests policy in January 2022 and request authors to consider both actual and perceived competing interests. Please review the policy

<https://www.embopress.org/competing-interests> and update your competing interests if necessary.

Please name this section 'Disclosure and Competing Interests Statement' and put it after the Acknowledgements section.

We declare no competing interests.

- The nomenclature of the EV figure legends in the manuscript text file is not correct: It should be 'Figure EVx' instead of 'Expanded View Figure X'.

We have corrected the nomenclature of the EV figures.

- Please use our reference format (last name, first name initial; et al should be used after the 10th name; DOIs should only be used for preprints and datasets that have not been published yet):

We have changed the reference format.

- Per journal policy, we do not allow 'data not shown', which is stated in the manuscript (page 19). All data referred to in the paper should be displayed in the main or Expanded View figures, or an Appendix. Thus, please add these data (or change the text accordingly if these data are not central to the study). See:

<https://www.embopress.org/page/journal/14693178/authorguide#unpublisheddata>

We thank the editor for this comment. The paragraph in question has been removed, and the text has been modified accordingly, as the data referred to were not central to the study.

- Please also make sure that each figure panel (main and EV figures) is called out separately and sequentially. Presently, there seem to be no separate callouts for panels 3F and 3G. Please check.

We have checked and did not find the issue mentioned below.

- Please check again that the number "n" for how many independent experiments were performed, their nature (biological versus technical replicates), the bars and error bars (e.g. SEM, SD) and the test used to calculate p-values is indicated in the respective figure legends. Please also check that all the p-values are explained in the legend, that exact p-values are listed and that these fit to those shown in the figure. Please provide statistical testing where applicable. Please avoid the phrase 'independent experiment' but clearly state if these were biological or technical replicates. Please also indicate (e.g. with n.s.) if testing was performed, but the differences are not significant. In case n=2, please show the data as separate datapoints without error bars and statistics. See also:

<http://www.embopress.org/page/journal/14693178/authorguide#statisticalanalysis>

If n<5, please show single datapoints for diagrams. Moreover:

- Please note that the exact p values are not provided in the legends of figures 2B, 4E, 5E, 6F, 7D, E.

- Please indicate the statistical test used for data analysis in the legends of figures 1C, D; 2A, B.

Figure legends now state *n*, replicate nature (biological vs technical), error bars (SD/SEM), statistical tests, and exact *P* values; "n.s." indicated where applicable.

- Please add to each legend (main, EV figures and Appendix Figures, where applicable) a 'Data Information' section explaining the statistics used or providing information regarding replicates and scales. See: <https://www.embopress.org/page/journal/14693178/authorguide#figureformat>

We have added a data information section in figure legend where applicable.

- Please make sure that all the funding information is also entered into the online submission system and that it is complete and similar to the one in the acknowledgement section of the manuscript text file. Presently, the grant numbers entered in the system (ECTZ158570, FDT202304016406, ECTZ60924, 2021-2-FJC-13113) are missing in the manuscript text file; the grant number 23515 mentioned in the Acknowledgements is missing in the submission system. Please check.

We have corrected the funding information.

- Please confirm that for all Western blot panels in the manuscript the loading control was run on the same gel as the other proteins detected. Please note that we discourage comparisons between samples on different gels/blots, even if the samples derive from one experiment, as confounding factors reduce comparability. If unavoidable, the figure legend must state that the samples derive from the same experiment and that gels/blots were processed in parallel. If a 'representative' loading control is shown for multiple gels/blots, the intra-gel controls should be shown in the source data files and the figure legends should describe the data displayed accurately. See our author guidelines: I confirm that for all Western blot panels in the manuscript the loading control is run on the same gel as the other proteins detected

- Please add the primer information (Table EV2) directly to the Reagents & Tools table. Please add callouts to the Reagents & Tools table where appropriate. Please also remove Table EV2 from the manuscript files.

We have introduced the primer information directly to the reagents and tool table.

- Please add the legend for Table EV1 on the first TAB of the excel sheet. Then remove the legends for EV tables from the main manuscript text file.

We have introduced the legend for table EV1 directly in the first TAB of the excel sheet.

- Thanks for providing the requested source data. However, please upload the source data as one folder per main figure, grouping together all the files (one for each panel) for this figure ZIPed together, and as one folder for the EV figures, grouping together all the files (one folder for each figure containing one file for each panel) ZIPed together.

Source data (Figure 1 and 5) have been reorganized and uploaded as ZIP folders

Dr. Clarisse Berlioz-Torrent
INSERM U1016, Institut Cochin
Cnrs, UMR8104, Univ. Paris Cité
27 rue du faubourg Saint Jacques
Paris 75014
France

Dear Clarisse,

I am very pleased to accept your manuscript for publication in the next available issue of EMBO reports. Thank you for your contribution to our journal.

Kind regards,

Martina
